# An AI-driven fire risk forecasting framework for urban villages using IGWO-optimized LSTM with incremental learning

Jiangxue Tian[1], Handong Li [2,3]*, Shuran Lv[1]

1 School of Management Engineering, Capital University of Economics and Business, Beijing, China, 2 School of Computing and Data Engineering, NingboTech University, Ningbo, China, 3 Department of Biochemical Engineering, University College London, London, United Kingdom

* handongli@nit.zju.edu.cn

## Abstract

Artificial intelligence (AI) is reshaping decision-support systems across multiple domains, including risk management and urban safety. Urban villages, characterized by high population density and informal infrastructure, are particularly vulnerable to fire hazards. This study presents an AI-driven fire risk forecasting framework based on an Improved Grey Wolf Optimizer (IGWO) and a Long Short-Term Memory (LSTM) neural network, further enhanced by an incremental learning strategy. IGWO improves hyperparameter convergence and avoids local optima, while the incremental component allows real-time model updates without full retraining. Using real fire incident data from 55 urban villages in Beijing, the proposed IGWO-LSTM-IL model achieves a 92.57% reduction in mean squared error compared to baseline LSTM. The model demonstrates high predictive accuracy, stability, and adaptability, making it a practical tool for intelligent fire risk monitoring and urban safety systems within the scope of AI-transforming urban infrastructure.

## 1 Introduction

Urban villages refer to areas located within or on the outskirts of a city that have not yet fully urbanized during the process of urban development. They possess some characteristics of both urban and rural areas, such as a lack of planning for building facilities [1], a large number of self built houses, dense population, and chaotic government management. These areas have been formed during the process of urbanization, preserving their original social and cultural characteristics. Urban villages are widely present worldwide, especially in rapidly developing cities in Asia and urbanized areas in Europe and America, reflecting the transitional state between urban expansion and rural traditions [2].

The structure and functional layout of urban villages have a high degree of spontaneity, and the diverse characteristics of different regions enhance the structural and

**Data availability statement:** All relevant data are within the manuscript and its Supporting Information files.

**Funding:** The author(s) received no specific funding for this work.

**Competing interests:** The authors have declared that no competing interests exist.

functional diversity of urban villages, making them play a unique role in the process of urbanization construction [3]. At the same time, they bring many safety hazards, such as dense buildings, complex personnel composition, and weak fire protection facilities. Specifically urban villages' self-built houses have even higher fire risk due to lack of formal design and regulation, using low standard of building material, etc [4]. Beside these, complex composition of residents, lack of safety awareness and necessary protective skills push the fire risk to a incredible high level [5]. As a result urban villages fire's casualties are more serious than those in cities. Therefore, carrying out research specifically on urban villages' fire risks benefits building out prediction scheme, and potential risks and accidents are more likely to be eliminated as a result of that [6].

Currently researches on fire risk prediction are mainly focuses on the fields of urban buildings and forest, and these researches have drawbacks of poor timeliness and strong subjectivity since they rely on on-site inspections and empirical formulas. For example, [7] presented an approach to generating a forest fire risk map by integrating geographic information system–based multiple criteria decision analysis (GIS-MCDA) with the analytic hierarchy process (AHP) and a statistical index (SI). In paper [8], it constructed a novel approach to predicting fire risk in buildings by leveraging advanced machine learning techniques and integrating diverse datasets. a fire prediction and evacuation system based on cellular automata for large buildings with dense population. Besides these, more studies focus on fire risks in special scenarios, such as goafs in coal mines [9], inside mines [10], and specific city districts [11], etc. However, the research on urban villages, which are areas with a high incidence of fires, is relatively scarce.

Although some scholars focus on multi-factor influences on urban fire risks, the disaster-causing factors of fires in urban villages are special, and are much more complex than those in urban areas. Conventional analysis methods such as AHP, Entropy Method and Markov model, etc. are difficult to be applied in urban village scenarios. For example, [12] carried out a hierarchical integrated spatial risk assessment in India's core city using GIS and AHP. This method has issues such as strong subjectivity and inability to handle time-series data.

Some researchers have explored more advanced prediction technologies like machine learning and big data analysis. [13] developed a deep learning model based on the UNet architecture to achieve fast and accurate prediction of fire temperature field under the ceiling in complex planar rooms. However, this model is limited to special scenarios and not suitable for general cases. [14] obtained forest fire susceptibility maps of the Babolrood Watershed in the Mazandaran Province of Iran from random forest, artificial neural network and logistic regression models. Although the random forest model used in this study performs well in terms of prediction accuracy, it does not have the ability to update in real time. The timeliness of the model is limited by the time range of data collection, thus it could not reveal the current fire risk status in real time. [15] designed three kinds of network models based on Long Short-Term Memory (LSTM) to predict fire spread rate, exploring the interaction between fire and wind. Recent studies have further demonstrated the effectiveness

of LSTM-based models in complex time-series prediction tasks. For example, Kong et al. [16] applied LSTM networks for short-term load forecasting, while Qin et al. [17] proposed a dual-stage attention-based recurrent neural network for time-series prediction. However, LSTM still suffers from issues such as long training time, sensitivity to hyperparameters, and potential overfitting.

Intelligent swarm optimization algorithm, as a machine learning algorithm, is commonly used in various decision support models due to its powerful optimization ability. At the same time, it can also optimize the parameters of neural networks to improve the performance of models. [18] optimized the hyperparameters in the eXtreme Gradient Boosting (XGBoost) model using Grey Wolf Optimizer (GWO) algorithm to create a fire growth rate warning map for the Liangshan Prefecture in Sichuan Province, China. [19] used a Neural Fuzzy inference system (NF) to establish the forest fire model whereas Particle Swarm Optimization (PSO) was adopted to investigate the best values for the model parameters. However, traditional intelligent swarm optimization algorithms have problems such as being prone to falling into local optima, slow convergence speed, and poor robustness. Recent studies have also emphasized the importance of evidence-based forecasting principles in predictive modeling. Armstrong and Green [20] proposed forecasting method checklists to improve the transparency and reliability of forecasting practices. In addition, large-scale forecasting evaluations such as the M4 competition [21] highlight the importance of rigorous model comparison in time-series forecasting research.

To solve the above issue, we propose a grey wolf optimizer (IGWO) by introducing nonlinear convergence factors and Gaussian mutation operators [22]. By optimizing the hyperparameters of LSTM using the proposed IGWO, a time-series data fire prediction model is constructed. LSTM is a special type of recurrent neural network (RNN) designed to solve the problems of gradient vanishing and exploding in traditional RNNs when processing long sequence data [23]. LSTM introduces input gates, forget gates, and output gates to control the flow of information, enabling it to learn and remember long-term dependency information and more effectively process and predict time-series data [24]. However, LSTM has shortcomings such as long training time, sensitivity to hyperparameters, and susceptibility to overfitting. Thus, an improved GWO (IGWO) algorithm is proposed to improve LSTM in this paper. The Grey Wolf Optimization Algorithm (GWO) is an optimization algorithm that simulates the social behavior and predation strategy of grey wolves [25]. Due to its simple structure, few parameters, and ease of adjustment, it is often applied to optimize the parameters of neural network models and usually exhibits good convergence speed and solution accuracy [26]. The proposed IGWO can further improve the performance of the algorithm by introducing dynamic weights and mutation operators to enhance the LSTM prediction model.

Based on the time-series data of fire-related factors in urban villages, the model can effectively process real-time data and accurately analyze the complex nonlinear relationships between fire influencing factors and fire probability in such areas. At the same time, the incremental learning mechanism enables the model to update in real time and adapt to environmental changes, so that the accuracy and practicality of fire risk diagnosis is significantly improved. This study provides a scientific basis and technical support for fire prevention and control in urban villages. The main contributions and advantages of the proposed method could be given as follows.

- An improved Grey Wolf Algorithm (IGWO) is proposed to solve the problems of premature convergence, poor robustness, and slow convergence speed in traditional intelligent swarm optimization algorithms.

- A urban village fire risk prediction model based on the Improved Grey Wolf Optimization Algorithm (IGWO) and Long Short Term Memory Network (LSTM) is constructed. The proposed model could realize fire risk prediction in urban villages based on time-series data.

- Incremental learning (IL) strategy is integrated to the model to continuously update the model and achieve real-time prediction of fire risk in urban villages.

- A full-scale fire experiment is conducted to analyze the fire risk and verify the effectiveness of the proposed method.

The rest of this paper is structured as follows. Sect [1] introduces the basic concepts and related work, including LSTM, the GWO improvement process, and the incremental learning algorithm. Sect [2] describes the construction of the fire risk prediction model based on the proposed IGWO improved LSTM and incremental learning, including the determination of fire risk indicators and the model's implementation details. Sect [3] presents the experimental setup and case study analysis, verifying the model's performance. Finally, Sect [4] concludes the achievements of the paper and discusses future research directions.

## 2 Related work

### 2.1 LSTM model

LSTM is a recurrent neural network that designed specifically to handle long-term dependencies in time series data effectively [27]. While the convolutional neural network (RNN) can address time sequence issues and retain short-term memory, it lacks the capability for long-distance time sequence retention [28]. Additionally, recurrent neural networks (RNNs) often grapple with the issue of gradient disappearance or divergence during the training process, a phenomenon that frequently precipitates challenges in achieving convergence or the emergence of suboptimal local solutions, thus posing certain constraints [29]. The LSTM neural network incorporates memory cells into its hidden neurons, which control the state of memory cells through three gating units, including the forget gate, input gate, and output gate, to discard and transfer information and update states, ultimately outputting information for the next time step [30]. Thus, LSTM is now able to solve the issues caused by gradient vanishing and gradient explosion since it has the long-term memory capability for time-series data.

The information flow inside an LSTM unit, as illustrated in Fig 1, proceeds through the following four steps:

**Step 1: Forget gate.** The forget gate determines what portion of the previous memory cell state $m_{t-1}$ should be retained. It uses the current input $x_t$ and the previous hidden state $s_{t-1}$:

$$f_t = \sigma \left( W_f \cdot [s_{t-1}, x_t] + b_f \right) \tag{1}$$

where $\sigma(\cdot)$ is the sigmoid activation function, $W_f$ is the weight matrix for the forget gate, and $b_f$ is the bias term.

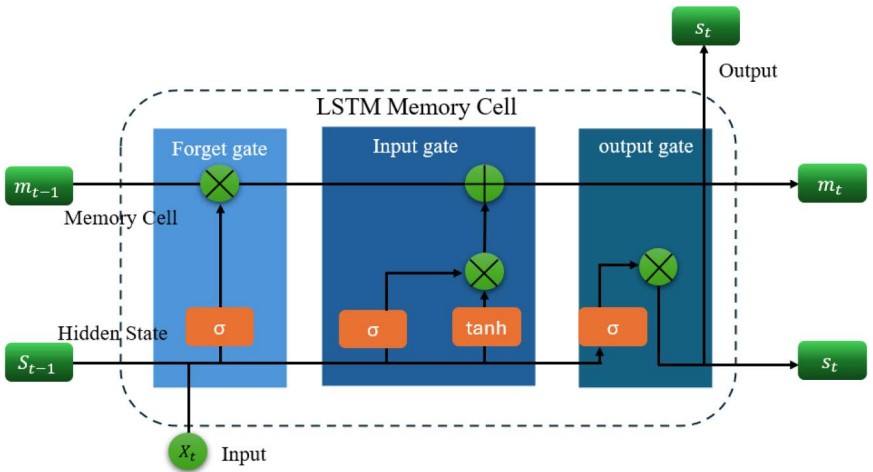

**Fig 1. Structure of the Long Short-Term Memory (LSTM) unit.** The architecture illustrates the memory cell $C_t$, hidden state $h_t$, and the three gating mechanisms (input gate, forget gate, and output gate) that regulate information flow within the LSTM network.

**Step 2: Input gate and candidate memory update.** The input gate determines how much new information is written to the memory cell. First, a candidate memory update $\tilde{m}_t$ is generated using the hyperbolic tangent function:

$$\tilde{m}_t = \tanh\left(W_c \cdot [s_{t-1}, x_t] + b_c\right) \tag{2}$$

Then, the input gate value $i_t$ is calculated as:

$$i_t = \sigma\left(W_i \cdot [s_{t-1}, x_t] + b_i\right) \tag{3}$$

**Step 3: Memory cell state update.** The current memory cell state $m_t$ is updated by combining the retained memory and the new candidate values:

$$m_t = f_t \cdot m_{t-1} + i_t \cdot \tilde{m}_t \tag{4}$$

**Step 4: Output gate and hidden state update.** The output gate determines the new hidden state $s_t$, based on the updated memory state:

$$o_t = \sigma\left(W_o \cdot [s_{t-1}, x_t] + b_o\right) \tag{5}$$

$$s_t = o_t \cdot \tanh(m_t) \tag{6}$$

This structure allows LSTM networks to selectively retain relevant temporal information and effectively model long-range dependencies. In the context of urban village fire risk prediction, the LSTM's ability to capture dynamic, nonlinear time-series patterns between fire risk indicators proves highly valuable. However, LSTM performance heavily depends on hyperparameter tuning, such as the learning rate, hidden units, dropout rate, and training epochs. Suboptimal settings may cause overfitting, slow convergence, or entrapment in local minima. To address this, we integrate an Improved Grey Wolf Optimizer (IGWO) to perform global search over the hyperparameter space, thereby enhancing LSTM prediction accuracy and stability.

## 2.2 Improved Grey Wolf Optimizer (IGWO)

The Grey Wolf Optimizer (GWO) is a metaheuristic inspired by the social hierarchy and hunting strategies of grey wolves [31]. The population is divided into four roles based on fitness: $\alpha$ (leader), $\beta$ (sub-leader), $\delta$ (third leader), and $\omega$ (follower). The optimization process is guided by the top three wolves, which simulate the encircling and attacking of prey, as illustrated in Fig 2.

At each iteration, the wolves update their positions based on the estimated location of the prey:

$$\mathbf{D} = \left| \mathbf{C} \cdot \mathbf{X}_p(t) - \mathbf{X}(t) \right| \tag{7}$$

$$\mathbf{X}(t+1) = \mathbf{X}_p(t) - \mathbf{A} \cdot \mathbf{D} \tag{8}$$

Here, $\mathbf{X}_p(t)$ is the prey's position (best solution), and $\mathbf{X}(t)$ is the current position. The vectors $\mathbf{A}$ and $\mathbf{C}$ are defined as:

$$\mathbf{A} = 2a \cdot \mathbf{r}_2 - a \tag{9}$$

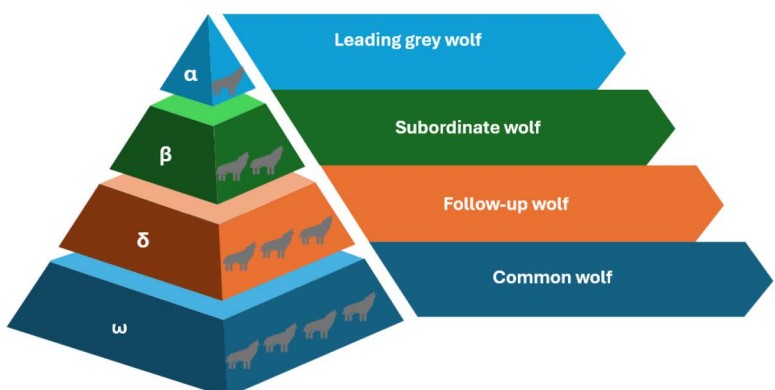

**Fig 2. Social hierarchy and hunting mechanism of the Grey Wolf Optimizer (GWO).** The population is divided into four levels (alpha, beta, delta, and omega), which collaboratively guide the search process toward the global optimum.

$$\mathbf{C} = 2 \cdot \mathbf{r}_1 \tag{10}$$

where $\mathbf{r}_1, \mathbf{r}_2 \sim \mathcal{U}(0, 1)$ are random vectors, and $a$ is the convergence factor linearly decreasing over iterations:

$$a = 2 - 2 \cdot \frac{t}{t_{\max}} \tag{11}$$

To improve search performance and avoid premature convergence, we propose two enhancements:

First, we replace the linear convergence factor with a nonlinear, sigmoid-based strategy to better balance exploration and exploitation:

$$\alpha(t) = \alpha_{\max} - \frac{1}{1 + \exp\left(-\frac{t}{t_{\max}}\right)} \tag{12}$$

The maximum convergence factor $\alpha_m ax = 2$ is set to the standard value in this stud.

Second, we modify the position update by simultaneously considering the three best wolves. The combined influence is computed as:

$$\mathbf{X}(t + 1) = \frac{1}{3} \sum_{i \in \{\alpha, \beta, \delta\}} \left( \mathbf{X}_i - \mathbf{A}_i \cdot \left| \mathbf{C}_i \cdot \mathbf{X}_i - \mathbf{X}(t) \right| \right) \tag{13}$$

To further enhance global search ability and maintain population diversity, we integrate a Gaussian mutation mechanism. Given the best solution $\mathbf{x} = (x_1, x_2, \ldots, x_n)$, a gene $x_m$ ($1 \le m \le n$) is randomly selected for mutation with probability $p_m$:

$$x'_m = \begin{cases} \max(\mathrm{lb}_m, \min(\mathrm{ub}_m, x_m + \delta)), & r < p_m \\ x_m, & \text{otherwise} \end{cases} \tag{14}$$

where

$$\delta = f_m \cdot (\mathrm{ub}_m - \mathrm{lb}_m) \cdot \epsilon, \quad \epsilon \sim \mathcal{N}(0, 1) \tag{15}$$

Here, $lb_m$ and $ub_m$ are the bounds for the $m$-th dimension, $f_m$ is the mutation intensity which set to 0.1 provides a moderate perturbation strength to balance exploration and convergence, and $\epsilon$ is a standard Gaussian noise term. The mutation probability $p_m$ is set to 0.1, which is a commonly used default value in evolutionary optimization algorithms to preserve population diversity while avoiding excessive randomness.

These modifications yield the Improved Grey Wolf Optimizer (IGWO), which achieves stronger exploration in early iterations and better convergence stability in complex, high-dimensional search spaces.

## 2.3 Benchmark evaluation and analysis

To evaluate the performance of the proposed Improved Grey Wolf Optimizer (IGWO), eight standard benchmark functions from the CEC2017 test suite [32] are employed in Fig 3. These functions represent a diverse set of characteristics, encompassing unimodal, multimodal, separable, and non-separable landscapes. Such diversity ensures that the simulation results are both comprehensive and scientifically robust. For comparative analysis, three widely recognized metaheuristic algorithms are selected: the original Grey Wolf Optimizer (GWO), Particle Swarm Optimization (PSO), and the Great Wall Construction Algorithm (GWCA), as in [33–35]. GWO, PSO, and GWCA are representative algorithms in metaheuristic algorithms, which have been widely applied and studied in academia and industry. This means that there are a large number of research and experimental results that can be used as references, which helps to make fair and comprehensive comparisons [36,37].

As shown in Fig 4, for the quadratic convex function $F_1$, IGWO quickly converges to the global minimum; When dealing with non convex functions $F_2$ and $F_3$, IGWO also exhibits fast and stable convergence characteristics; For the discontinuous function $F_4$ with a large number of local minima, the IGWO algorithm can effectively avoid getting stuck in local minima and quickly find the global optimal solution; When dealing with polynomial function $F_5$ with random noise, IGWO exhibits good robustness, although slightly fluctuating, it can still converge to the extremum; For complex functions $F_6$, $F_7$, and $F_8$ containing trigonometric functions, IGWO can effectively handle the non convexity and complexity of the functions, and quickly reach a stable state. These results indicate that the IGWO algorithm has significant advantages over traditional GWO, PSO, and GWCA algorithms in terms of global search capability, convergence speed, adaptability, and robustness. Especially when dealing with non convex function optimization problems with a large number of local minima and complex patterns, its performance is significantly better than other algorithms.

To balance exploration capability with computational efficiency, the population size for all algorithms is fixed at 10, and the maximum number of iterations is set to 500. Each algorithm is independently executed 30 times to assess convergence stability and robustness under stochastic initialization. The average and standard deviation of the optimal fitness values obtained across these runs are reported for each test function. A detailed overview of the benchmark functions is provided in Table 1, while their search space visualizations and convergence behaviors are depicted in Fig 3 and Fig 4, respectively. To better illustrate comparative performance, Fig 5 and Fig 6 present normalized heatmaps for the 30-dimensional and lower-dimensional (3D and 10D) cases. From the visual analysis in Fig 5, IGWO consistently achieves the lowest normalized average error across most 30-dimensional test functions, highlighting its superior convergence accuracy and stability compared to traditional GWO, PSO, and GWCA. Similarly, Fig 6 confirms that IGWO maintains its optimization effectiveness in lower-dimensional spaces, with particularly strong performance on precision-oriented functions such as $F_1$ through $F_4$.

## 2.4 Incremental learning strategy

Traditional batch learning algorithms operate under the assumption that the complete training dataset is available prior to model training. Once training is completed, the model ceases to learn, and no further updates are incorporated. However, in real-world scenarios, data is typically collected over time rather than in a single batch. Furthermore, the underlying

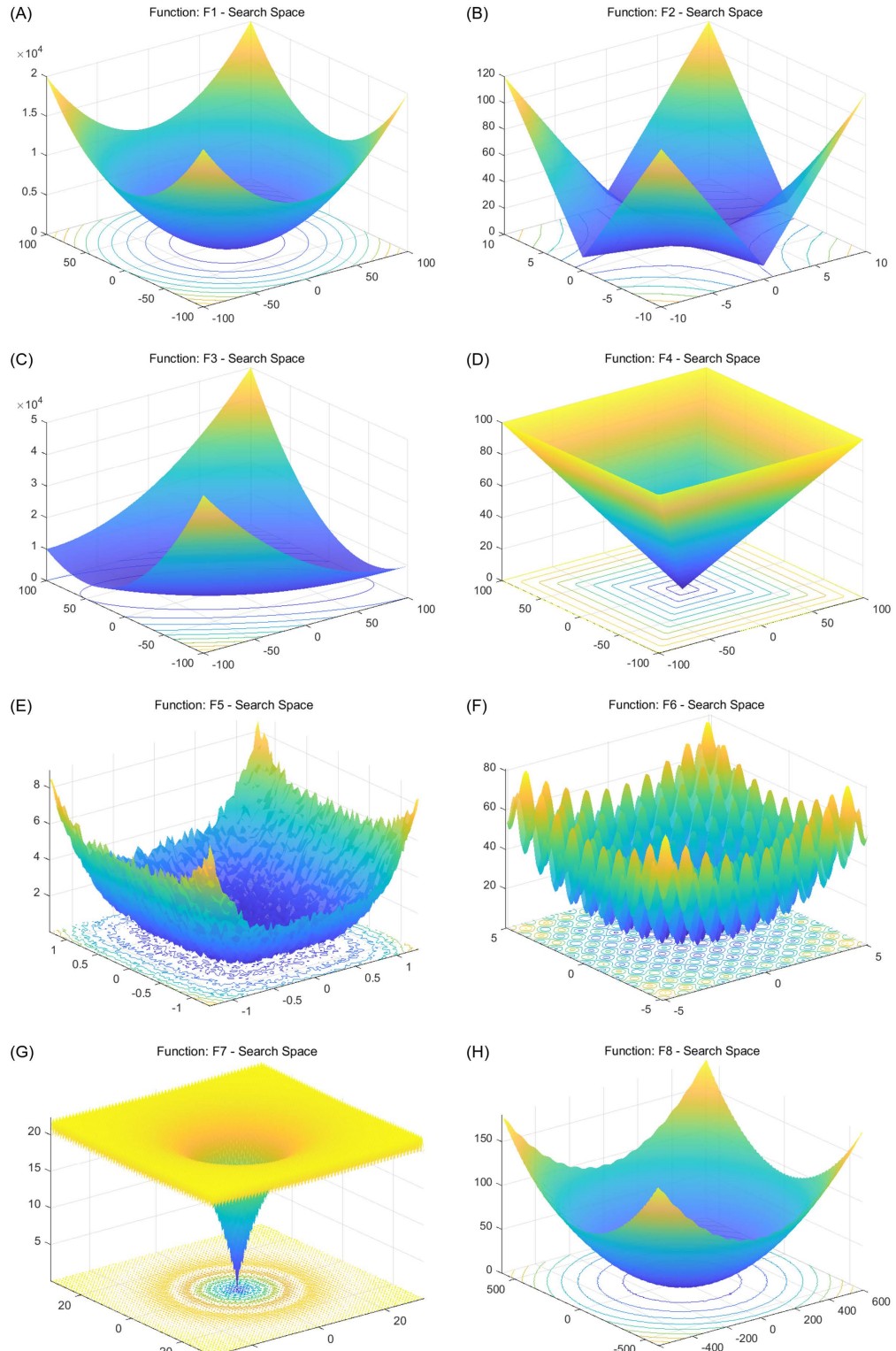

**Fig 3. Search landscapes of eight benchmark functions used to evaluate the optimization algorithms. (a)** Sphere ($F_1$); **(b)** Schwefel 2.22 ($F_2$); **(c)** Schwefel 1.2 ($F_3$); **(d)** Schwefel 2.21 ($F_4$); **(e)** Quartic with noise ($F_5$); **(f)** Rastrigin ($F_6$); **(g)** Ackley ($F_7$); **(h)** Griewank ($F_8$).

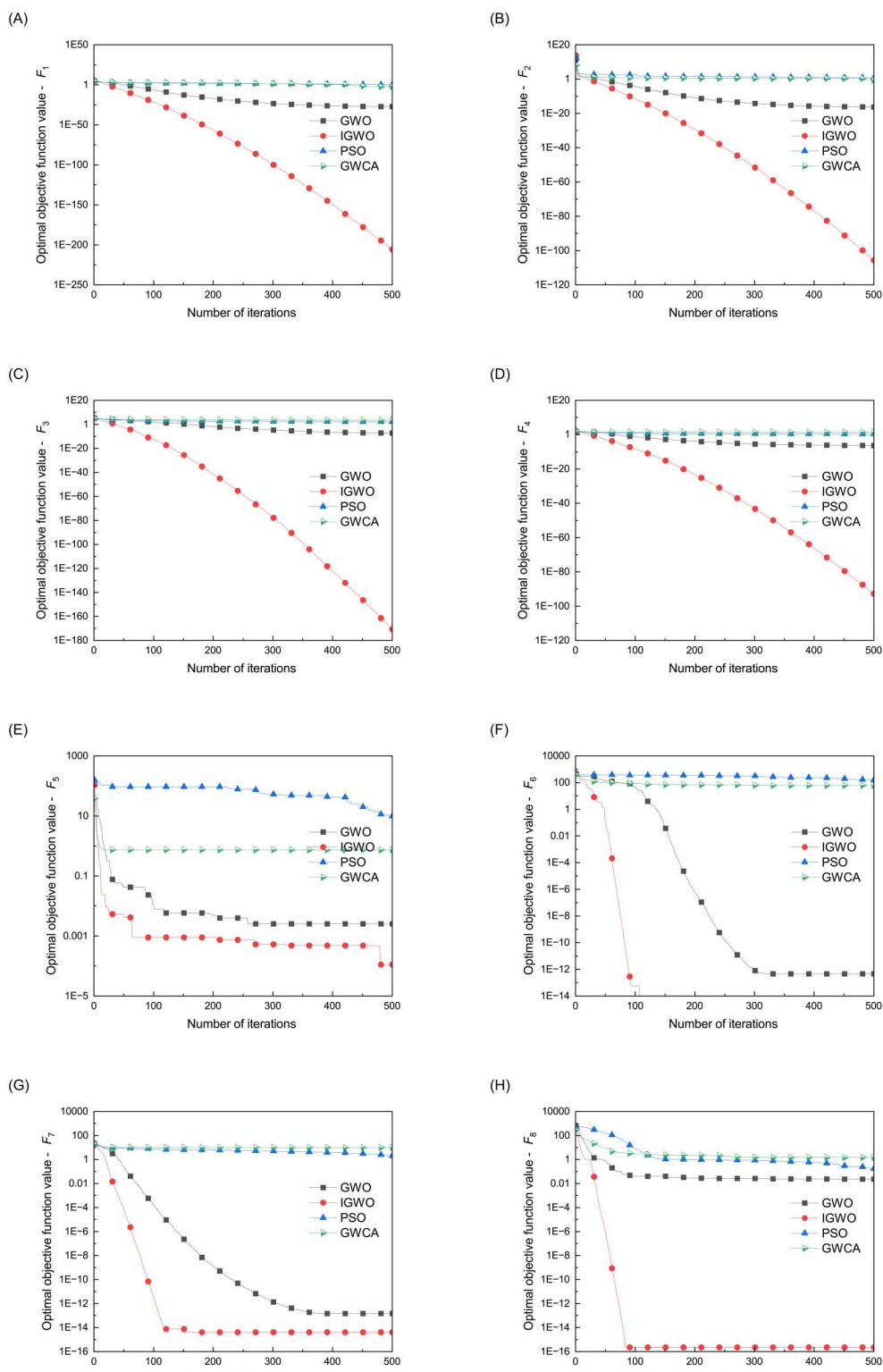

**Fig 4. Convergence curves of four optimization algorithms on the benchmark functions.** (a) $F_1$: Sphere; (b) $F_2$: Schwefel 2.22; (c) $F_3$: Schwefel 1.2; (d) $F_4$: Schwefel 2.21; (e) $F_5$: Quartic (with noise); (f) $F_6$: Rastrigin; (g) $F_7$: Ackley; (h) $F_8$: Griewank.

**Table 1. Definitions and properties of the benchmark functions used in this study.**

| No. | Name | Function Definition | Properties |
|---|---|---|---|
| $F_1$ | Sphere | $f_1(x) = \sum_{i=1}^{d} x_i^2$ | $d = 30$, $x_i \in [-100, 100]$, $f_{min} = 0$ |
| $F_2$ | Schwefel 2.22 | $f_2(x) = \sum_{i=1}^{d} |x_i| + \prod_{i=1}^{d} |x_i|$ | $d = 30$, $x_i \in [-10, 10]$, $f_{min} = 0$ |
| $F_3$ | Schwefel 1.2 | $f_3(x) = \sum_{i=1}^{d} \left( \sum_{j=1}^{i} x_j \right)^2$ | $d = 30$, $x_i \in [-100, 100]$, $f_{min} = 0$ |
| $F_4$ | Schwefel 2.21 | $f_4(x) = \max_i \{|x_i|\}$ | $d = 30$, $x_i \in [-100, 100]$, $f_{min} = 0$ |
| $F_5$ | Quartic (with noise) | $f_5(x) = \sum_{i=1}^{d} i \cdot x_i^4 + \text{random}[0, 1]$ | $d = 30$, $x_i \in [-1.28, 1.28]$, $f_{min} \approx 0$ |
| $F_6$ | Rastrigin | $f_6(x) = \sum_{i=1}^{d} \left[ x_i^2 - 10\cos(2\pi x_i) + 10 \right]$ | $d = 30$, $x_i \in [-5.12, 5.12]$, $f_{min} = 0$ |
| $F_7$ | Ackley | $f_7(x) = -20 \exp\left( -0.2\sqrt{\frac{1}{d}\sum x_i^2} \right)$ $- \exp\left( \frac{1}{d}\sum \cos(2\pi x_i) \right) + 20 + e$ | $d = 30$, $x_i \in [-32, 32]$, $f_{min} = 0$ |
| $F_8$ | Griewank | $f_8(x) = 1 + \frac{1}{4000}\sum x_i^2$ $- \prod \cos\left( \frac{x_i}{\sqrt{i}} \right)$ | $d = 30$, $x_i \in [-600, 600]$, $f_{min} = 0$ |

Note: All functions are standard benchmark functions commonly used in optimization algorithm evaluation.

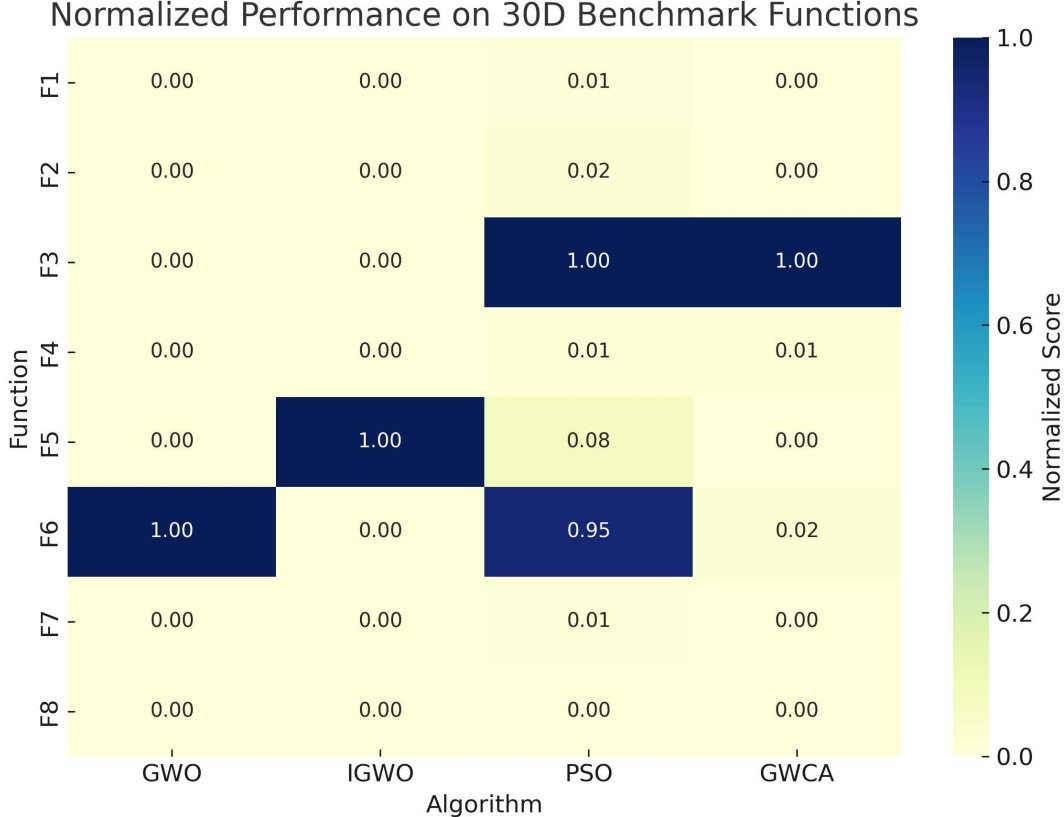

**Fig 5. Normalized performance comparison of the optimization algorithms on 30-dimensional benchmark functions.**

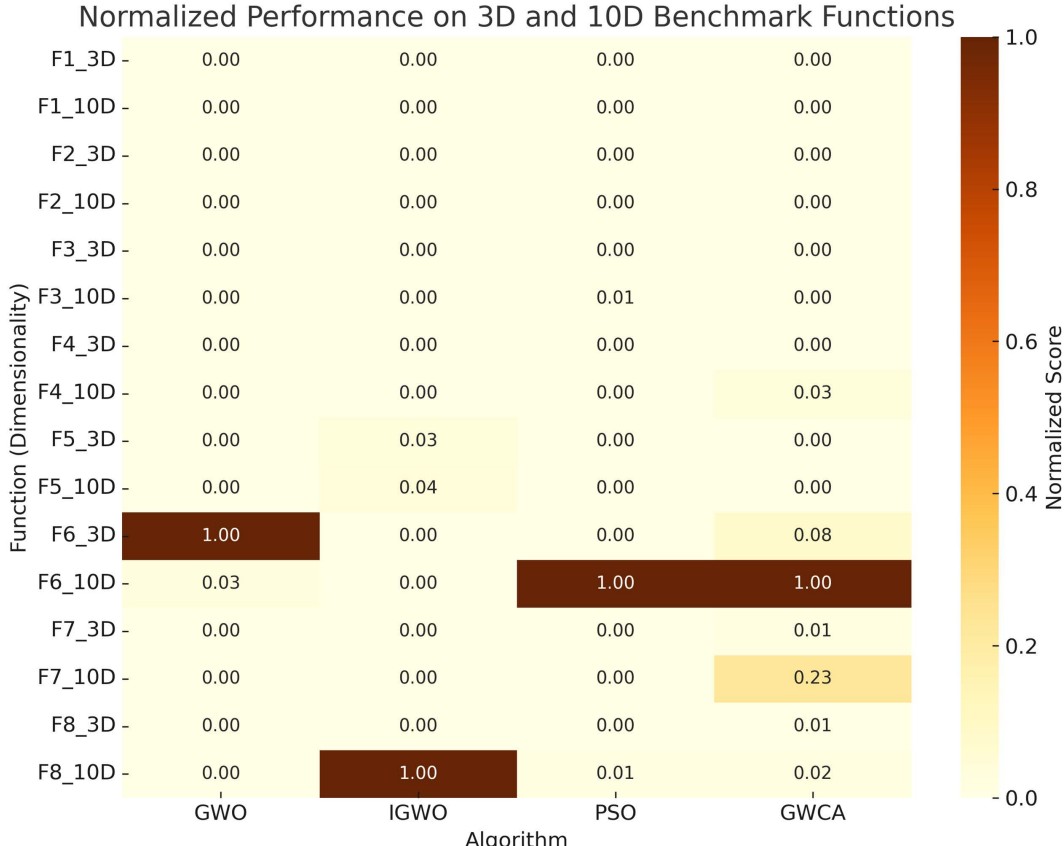

**Fig 6. Normalized performance comparison of the optimization algorithms on 3-dimensional and 10-dimensional benchmark functions.**

distribution or context of the data may evolve dynamically [38]. In such settings, re-training the entire model from scratch with each new data instance is computationally inefficient and impractical, especially for time-sensitive applications.

To address this challenge, incremental learning algorithms have been proposed. These methods enable continuous learning by updating model parameters as new data arrives. This process allows the model to revise, reinforce, and extend previously acquired knowledge without the need to access or retrain on the full historical dataset [39]. In this study, to enhance the real-time adaptability and responsiveness of the fire risk prediction model for urban villages, we integrate the optimized LSTM model with an incremental learning mechanism. This integration facilitates dynamic updates to the model as new fire-related data becomes available, ensuring timely adaptation to changing risk factors.

The proposed incremental learning strategy proceeds as follows:

1. **Data Acquisition:** Collect new characteristic data related to fire hazards in urban villages in real-time.

2. **Model Prediction and Error Calculation:** Feed the newly acquired data into the optimized LSTM model to compute the prediction output and the corresponding error.

3. **Parameter Update:** Fine-tune the model's weights and biases using the computed error via gradient-based optimization.

4. **Validation:** Evaluate the updated model on a validation set to ensure that its predictive performance remains accurate and stable.

The core of the incremental learning approach lies in efficient parameter adjustment. This work adopts an online gradient descent strategy, where the model parameters $\theta$ are iteratively updated based on the prediction loss. The update rule is defined as:

$$\theta_{t+1} = \theta_t - \eta \cdot \nabla_\theta \mathcal{L}(y_t, \hat{y}_t) \tag{16}$$

$$\mathcal{L}(y_t, \hat{y}_t) = \frac{1}{2} \|y_t - \hat{y}_t\|^2 \tag{17}$$

where $\theta_t$ denotes the model parameters at iteration $t$, $\eta$ is the learning rate, $y_t$ is the true value, $\hat{y}_t$ is the predicted value, and $\nabla_\theta$ represents the gradient operator with respect to the model parameters.

This strategy allows the model to quickly adapt to new conditions in urban fire risk environments, ensuring that predictions remain accurate and up-to-date without incurring the computational cost of full retraining. Specifically, in this study,the main settings are as the Table 2 as: (1) Learning rate: A small constant learning rate (1e-4) is used during incremental updates to ensure stable parameter adaptation. (2) Update frequency: Model updates are performed in a mini-batch manner when new data become available. (3) Drift detection: A performance-based drift detection mechanism is introduced. When the prediction error (RMSE) on recent data exceeds 110% of the historical average, the model update is triggered. (4) Catastrophic forgetting mitigation: To alleviate forgetting, a small portion of historical samples is retained and jointly used with new data during fine-tuning.

## 3 Proposed algorithm methodology

### 3.1 IGWO-LSTM-IL model

The integration of the Improved Grey Wolf Optimizer (IGWO) into the LSTM optimization framework is centered on leveraging IGWO's global search capability to automatically fine-tune key hyperparameters of the LSTM model—specifically, the number of hidden units, learning rate, and training epochs. Within the IGWO-LSTM architecture, the optimizer is designed to minimize the prediction error of the LSTM on a validation set, thereby enhancing model accuracy and

**Table 2. Configuration of the incremental learning framework.**

| Component | Setting/Description |
|---|---|
| Incremental learning strategy | Fine-tuning of a pre-trained LSTM without reinitialization |
| Learning rate | $1 \times 10^{-4}$, kept constant during incremental updates |
| Optimizer | Adam |
| Update frequency | Mini-batch updates performed when new data arrive |
| Batch size | 10 |
| Number of epochs | 5–10 for each incremental update |
| Data used for update | Newly arriving data together with a small portion of historical data through a replay mechanism |
| Replay ratio | Approximately 20% of historical samples |
| Drift detection method | Rolling RMSE-based monitoring |
| Drift threshold | Model update triggered when RMSE exceeds 110% of the historical average |
| Validation strategy | Online evaluation using recent samples |
| Catastrophic forgetting | Alleviated by replaying historical samples during fine-tuning |
| Model initialization | Continued from previously trained model parameters |

robustness while significantly reducing manual trial-and-error in hyperparameter tuning. Given the complex and nonlinear nature of fire risk factors in urban villages, conventional LSTM models often suffer from limitations such as slow convergence, gradient vanishing/explosion, overfitting, and susceptibility to local optima. By contrast, the IGWO-based optimization approach facilitates an efficient global exploration of the hyperparameter space, enabling the discovery of near-optimal configurations in fewer iterations.

The optimization process begins by initializing a population of grey wolves, where each wolf encodes a unique LSTM hyperparameter configuration. The fitness of each individual is evaluated based on the LSTM model's prediction error. The positions of the wolves are then iteratively updated according to IGWO's hierarchical hunting mechanism, in which the top-ranked wolves ($\alpha$, $\beta$, and $\delta$) guide the others toward better solutions. Once the optimal set of hyperparameters is identified, it is used to train the final LSTM model. The overall training and update process of the proposed IGWO-LSTM-IL model is illustrated in Fig 7, while the corresponding implementation logic is outlined in Algorithm 1. The workflow begins with the construction of the LSTM network, followed by hyperparameter optimization using the Improved Grey Wolf Optimizer (IGWO). Once the optimal configuration is identified, the LSTM model is trained and incrementally updated in real-time using newly arriving data. This hybrid approach ensures that the model continuously adapts to dynamic fire risk patterns while maintaining high prediction accuracy and computational efficiency.

**Algorithm 1. IGWO-LSTM Training with Incremental Learning.**

```
    Input: Initial LSTM network, max iterations T, data stream D
    Output: Trained and updated IGWO-LSTM model
1   Initialize wolf pack with random hyperparameters;
2   for t=1 to T do
3   │  Evaluate fitness of each wolf (LSTM model accuracy);
4   │  Update positions of α, β, δ;
5   └  Apply mutation operation to enhance diversity;
6   Assign best wolf's parameters to LSTM and retrain;
7   while new data d_t ∈ D arrives do
8   │  Compute prediction ŷ_t;
9   │  Calculate error L(y_t, ŷ_t) = ½‖y_t − ŷ_t‖²;
10  │  Update weights: θ_{t+1} = θ_t − η · ∇_θ L;
11  └  Validate model and monitor drift;
12  return final IGWO-LSTM model;
```

The proposed prediction framework is built upon a five-layer LSTM network, which includes: an input layer, an LSTM layer, a dropout layer, a fully connected (dense) layer, and a regression output layer. The input layer receives multivariate time-series data representing fire risk factors in urban villages. The LSTM layer captures temporal dependencies and dynamic patterns through gated recurrent units. To prevent overfitting, the dropout layer randomly disables a portion of neurons during training. The fully connected layer transforms the LSTM outputs to match the dimensionality of the target variable, while the regression layer computes the prediction loss and enables backpropagation during training.

To optimize the LSTM model's performance, the Improved Grey Wolf Optimizer (IGWO) is employed to tune three key hyperparameters: the number of hidden units, learning rate, and number of training epochs. Each grey wolf represents a unique combination of these hyperparameters, encoded as a position vector in a three-dimensional search space. These wolf individuals are initialized randomly within predefined bounds.

The fitness of each wolf is evaluated using the root mean square error (RMSE) between the predicted and actual values on the training dataset:

$$\text{RMSE} = \sqrt{\frac{1}{n} \sum_{t=1}^{n} \left(x_t - \hat{x}_t\right)^2}$$

(18)

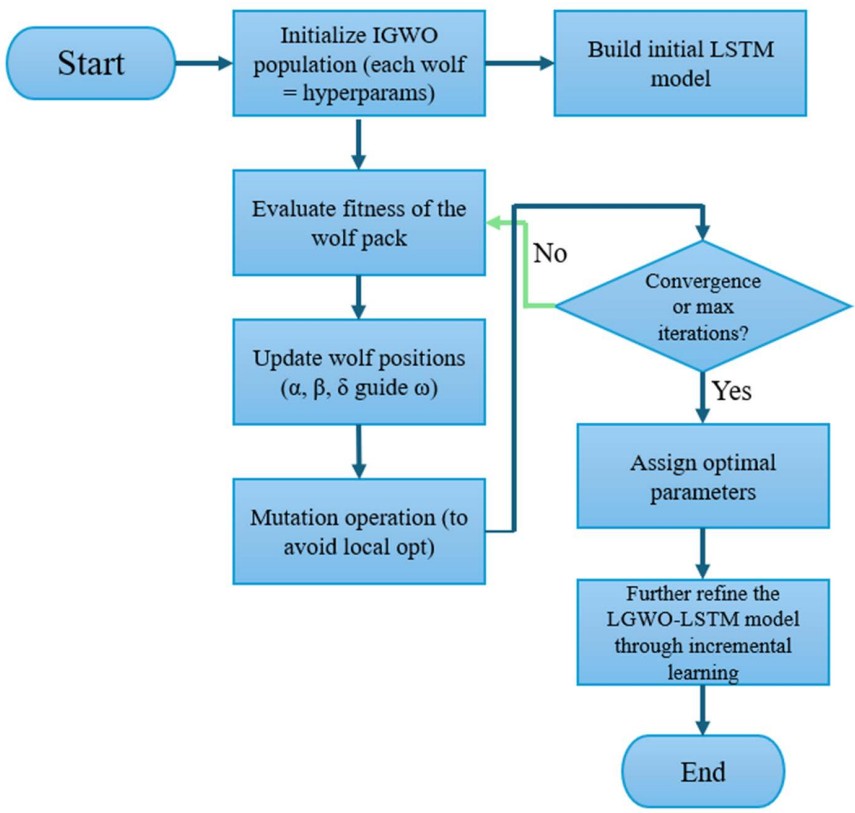

**Fig 7. Flowchart of the proposed IGWO-LSTM-IL fire risk prediction framework.** The workflow includes data preprocessing, Bayesian network-based fire probability estimation, parameter optimization using the Improved Grey Wolf Optimizer (IGWO), LSTM model training, and incremental learning for model updating.

where $n = N_o$ denotes the number of samples, $x_t$ is the expected output, and $\hat{x}_t$ is the predicted value of the model.

Through iterative optimization guided by the hierarchy of $\alpha$, $\beta$, and $\delta$ wolves, IGWO converges to the optimal hyper-parameter configuration. This configuration is then used to construct and train the final LSTM model. Once the model achieves satisfactory accuracy on the training set, its performance is evaluated on the testing dataset. To enable real-time adaptation to newly available fire risk data, the model is further refined using an incremental learning strategy. New input samples are standardized and used to update the trained model via online learning, without requiring complete retraining. This strategy not only maintains computational efficiency but also ensures model relevance in dynamic urban environments.

In summary, the IGWO-LSTM model demonstrates strong capabilities in processing time series data and capturing complex interactions between fire risk variables. When integrated with incremental learning, it becomes a robust and adaptive tool for predicting fire risk in real-time in urban village settings.

## 3.2 Determination of fire risk factor indicators

The causes of fires in urban villages are multifaceted, involving a combination of environmental, demographic, and management-related factors. To systematically identify relevant fire risk indicators for urban villages, we consulted multiple authoritative sources, including the *China Fire and Rescue Yearbook*, government-issued fire accident investigation reports, and research publications on urban village fire incidents from academic and institutional sources [40].

In addition, on-site field investigations were conducted in collaboration with local fire safety authorities, including the Chaoyang District Fire and Rescue Detachment (2024). A statistical analysis of reported fire incidents over the past five years was also performed to inform indicator selection.The fire risk indicators in this study were constructed to focus on ignition-related (hazard-triggering) factor within urban village environments, based on statistical analysis of 101 real fire incidents in Beijing. The final set of fire risk factors identified for suburban towns is summarized in Table 3.

### 3.3 Risk Prediction and analysis

It should be noted that two independent datasets are used in this study for different purposes. The first dataset consists of real fire incident records collected from 55 urban villages in Chaoyang District during 2023, including 101 fire events. These data are used to statistically analyze the distribution of fire causes and to construct the Bayesian network. The second dataset is a questionnaire-based time-series dataset obtained from 100 representative urban villages over a 30-day observation period. Therefore, the dataset provides 3,000 temporal observations. Expert scoring is applied to evaluate the fire risk indicators, and the Bayesian network is used to generate the corresponding fire probability, which serves as the target variable for training the prediction model. To clarify the overall workflow of the proposed method, the complete data processing and modeling framework is illustrated in Fig 8.

Based on field surveys conducted in collaboration with local government departments, including the Chaoyang District Fire and Rescue Detachment (2024), a total of 101 fire incidents were recorded across 55 urban villages in Chaoyang District during the year 2023. This section analyzes the statistical distribution of fire causes and assesses the predictive performance of the proposed model on these real-world data.

Based on expert opinions, relevant literature [41–43], and local government documents, 12 fire risk factors in urban villages were identified. According to the statistical analysis of 101 fires in urban villages in Chaoyang District, the number of fires caused by each fire factor was obtained, as shown in Table 4.

To quantify the intermediate nodes $A_1$, $A_2$, $A_3$, and $A_4$ in the Bayesian network shown in Fig 9, conditional probabilities are computed based on the states of their corresponding parent nodes $B$. For instance, node $A_1$, representing fire risks related to electrical systems, is influenced by four parent factors: electrical circuit faults ($B_1$), electrical equipment faults or improper use ($B_2$), electric vehicle faults or improper use ($B_3$), and fuel vehicle faults or improper operation ($B_4$).

The weight of each parent node $B_i \in \mathcal{B}_j$ contributing to an intermediate node $A_j$ is defined by:

**Table 3. Fire Risk Factors for Suburban Towns.**

| First-Level Indicator | Second-Level Indicator |
| --- | --- |
| Electricity | Electrical circuit failure |
| | Electrical equipment failure or improper use |
| | Electric vehicle failure or improper use |
| | Fuel vehicle failure or improper operation |
| Industrial Operations | Improper production operation |
| | Careless storage |
| | Production equipment failure |
| | Re-ignition of residual fire |
| Careless Use of Fire in Daily Life | Lighting or stove failure/improper use |
| | Careless burning of waste, outdoor fires |
| Others | Arson |
| | Spontaneous combustion, lightning strike, etc. |

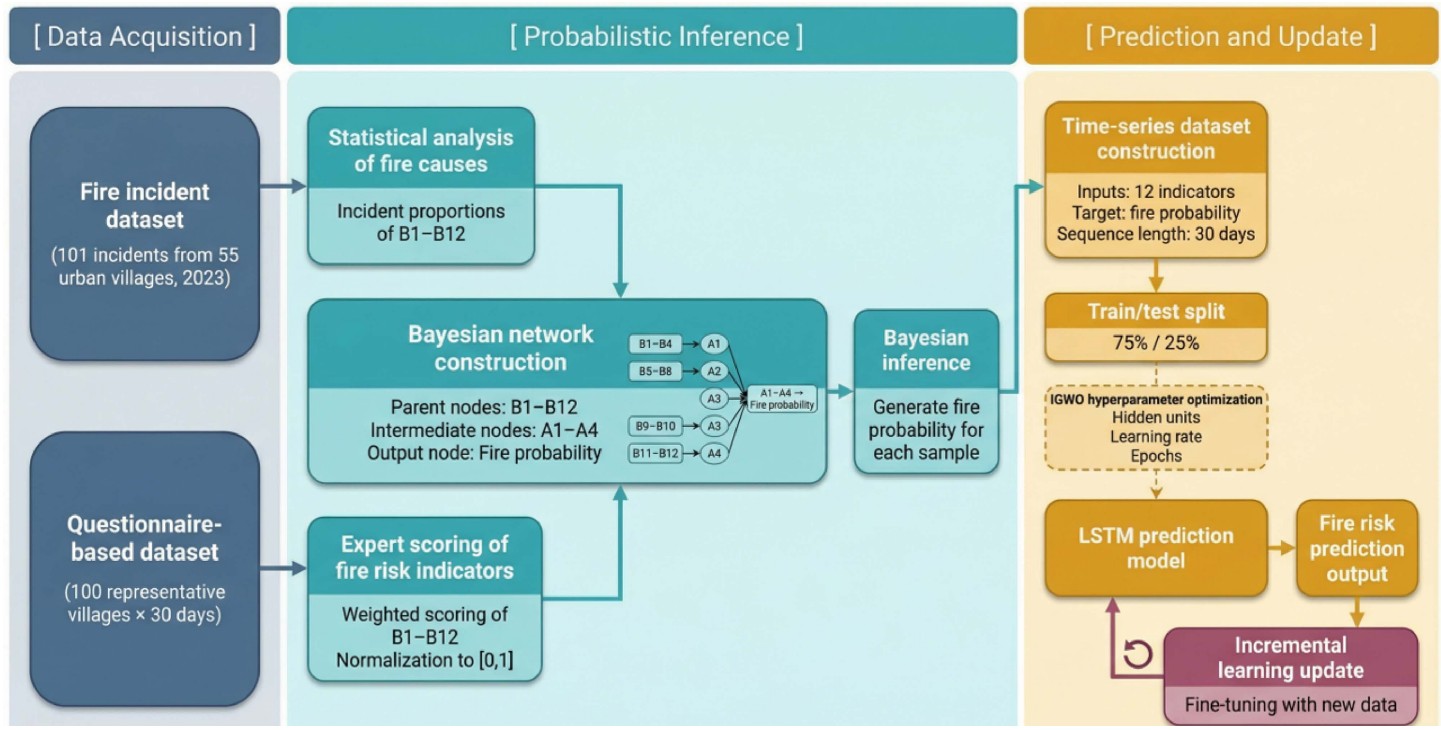

**Fig 8. Data processing pipeline used in the study.** The pipeline illustrates the transformation from expert questionnaire data and fire risk indicators to Bayesian probability estimation and the subsequent preparation of time-series inputs for the IGWO-LSTM-IL prediction model.

$$\omega'_{Bi} = \frac{\omega_{Bi}}{\sum_{k\in\mathcal{B}_j} \omega_{Bk}}, \quad \forall B_i \in \mathcal{B}_j \tag{19}$$

where $\mathcal{B}_j$ is the set of parent nodes for $A_j$, and $\omega_{Bi}$ is the original proportion of fire cause $B_i$ from the statistics.

If the state (or score) of node $B_i$ is denoted as $a_{Bi} \in [0, 1]$, the probability of node $A_j$ is computed as:

$$P_{Aj} = \sum_{i\in\mathcal{B}_j} \omega'_{Bi} \cdot a_{Bi} \tag{20}$$

The final fire probability is calculated by aggregating the contributions from all intermediate nodes $A_j$, weighted by their total proportions:

$$\omega_{Aj} = \sum_{i\in\mathcal{B}_j} \omega_{Bi} \tag{21}$$

$$P = \sum_{j=1}^{4} \omega_{Aj} \cdot P_{Aj} \tag{22}$$

**Table 4. Statistics of Fire Risk Factors in Urban Villages.**

| Cause Code | Number of Incidents | Proportion |
|---|---|---|
| B1 | 15 | 0.1485 |
| B2 | 8 | 0.0792 |
| B3 | 24 | 0.2376 |
| B4 | 11 | 0.1089 |
| B5 | 3 | 0.0297 |
| B6 | 9 | 0.0891 |
| B7 | 3 | 0.0297 |
| B8 | 16 | 0.1584 |
| B9 | 2 | 0.0198 |
| B10 | 8 | 0.0792 |
| B11 | 1 | 0.0099 |
| B12 | 1 | 0.0099 |

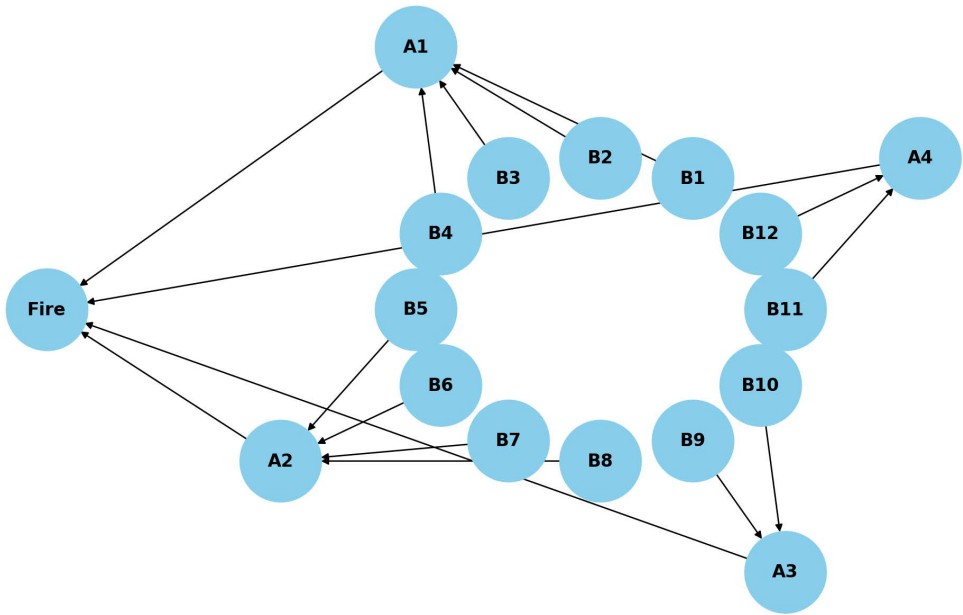

**Fig 9. Bayesian network representing the relationships between fire causes and urban village fire risk categories.**

## 4 Case study

According to recent research conducted by the Beijing Municipal Institute of City Planning and Design, there are currently 501 urban villages in Beijing, predominantly located between the 5th and 6th ring roads. In a targeted study conducted over a period of 30 days, 100 representative urban villages were selected. Six experienced experts in fire prevention and control assessed fire risk indicators through structured questionnaires. Participation in the questionnaire survey was voluntary, and informed consent was obtained from all participants prior to participation. The questionnaire collected only professional assessments of fire risk indicators and did not involve personal or sensitive information. Therefore, formal ethical approval was not required for this study. So, a fuzzy–Bayesian integrated framework was adopted to reduce subjectivity

in expert judgment. Experts from relevant fields (fire rescue, emergency management, architecture, and safety science) evaluated each fire risk indicator using a 0–100 scale. Their assessments were mapped to triangular fuzzy numbers corresponding to five linguistic levels (Very Low to Very High). Expert weights (0.2 for fire rescue and emergency management experts, 0.1 for the architecture expert, and 0.3 for the senior safety science professor) were assigned according to domain relevance and experience. The weighted fuzzy assessments were aggregated using $\lambda$-cut sets ($\lambda = 0.1$) and defuzzified using the integral value method to obtain crisp scores ($a_{B_i}$) within the interval [0,1]. These scores were then used as inputs to the Bayesian network. The detailed questionnaire design, representative example data, and implementation code used in this study are available at: https://github.com/handongli2019/Urban-Village-Fire-Risk-Prediction-Using-IGWO-Optimized-LSTM-with-Incremental-Learning. The scores were then organized into 100 samples, each with a 30-day time step and 12 fire risk feature inputs for the IGWO-LSTM-IL model. The output mode is set to "last," meaning each sample outputs the fire probability at the final time step derived from the Bayesian network. The dataset is divided into training and testing sets for model development and validation. Samples 1–75 are used for training and samples 76–100. All reported results are based on strict out-of-sample testing (75/25 train/test split) to ensure the model's predictive validity and generalization ability. The random seed is set to 42 to ensure reproducibility. To evaluate the practical performance of the proposed fire risk prediction model, this case study was conducted using the fire probability outputs derived from the Bayesian network in conjunction with the IGWO-LSTM-IL model.

To establish a baseline for fire risk prediction in urban village environments, a Long Short-Term Memory (LSTM) network was trained on the dataset without optimization by metaheuristic algorithms. The performance of this model is illustrated in Fig 10 and Fig 11, which highlight the prediction error characteristics and the correspondence between predicted and actual risk values, respectively. Fig 10 presents the distribution of prediction errors. The histogram demonstrates a generally symmetric distribution centered near zero, with a slight left skew (skewness = −0.524), indicating that the model occasionally underestimates fire risk. The superimposed normal distribution curve confirms that the residuals approximate a Gaussian distribution, with a kurtosis of approximately 0.007, suggesting neither pronounced peaks nor heavy tails. The standard deviation of the error is 0.146, and the mean error is −0.050, implying a minor overall overestimation by the model. Fig 11 compares the predicted and actual fire risk values across the sample index. The shaded area represents the error band, defined as ± the absolute error around the predicted value. The visual alignment between the predicted and actual curves confirms the model's ability to capture the temporal dynamics of risk variation, although the width of the error band in some segments highlights regions with higher uncertainty.

Overall, the LSTM baseline provides reasonable predictive capacity with moderate variance in error. However, the observed bias and residual spread suggest potential for improvement, which motivates the introduction of enhanced models optimized by intelligent algorithms in subsequent sections.

In the simulation, both the Grey Wolf Optimizer (GWO) and the Improved Grey Wolf Optimizer (IGWO) have a population size of 30, with a maximum iteration count of 10, and operate in a 3-dimensional search space. A single-layer LSTM network is adopted in this study, with the number of hidden layers fixed to one. It should be noted that LSTM performance is highly sensitive to hyperparameter settings. While metaheuristic algorithms such as PSO or GWO can theoretically be used to optimize LSTM weights, such an approach would significantly increase the computational burden due to the high dimensionality of the weight space. Therefore, in this study, the IGWO algorithm is employed only for hyperparameter optimization, while the LSTM weights are trained using standard gradient-based backpropagation to ensure computational efficiency and stable convergence.The key LSTM hyperparameters—number of hidden units, number of training epochs, and learning rate—are optimized using the GWO and IGWO algorithms. The batch size is set to 32 based on the dataset characteristics (number of features, time steps, and samples). To ensure robustness, each model was independently executed 30 times with different random initializations, and the average results are reported as the final prediction outcomes. The search space of LSTM hyperparameters was defined as follows: the number of hidden units was set within [16, 128], the learning rate within [0.0001, 0.01], and the number of epochs within [50, 200]. These ranges were selected based on

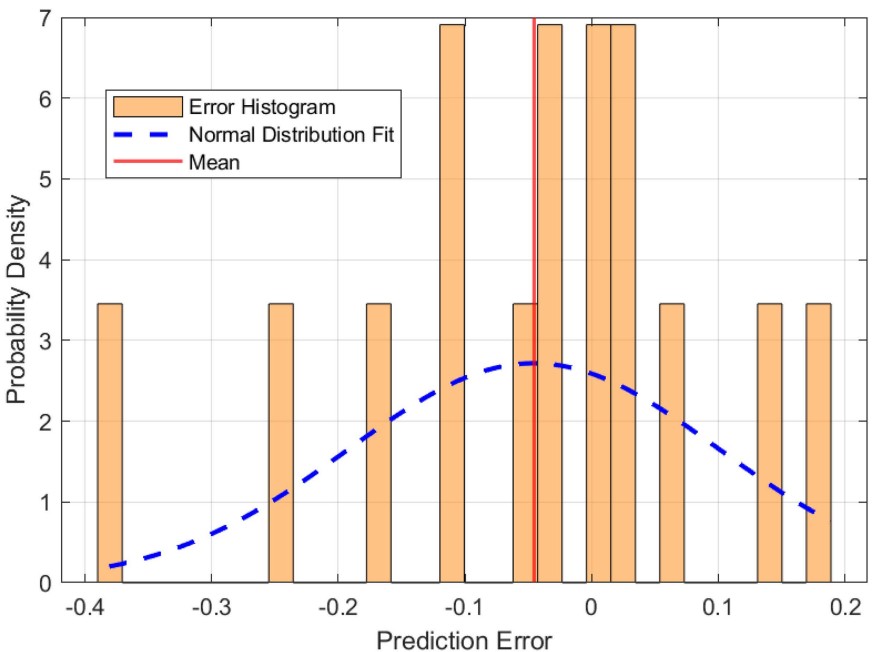

**Fig 10. Distribution of prediction errors for the baseline LSTM model.**

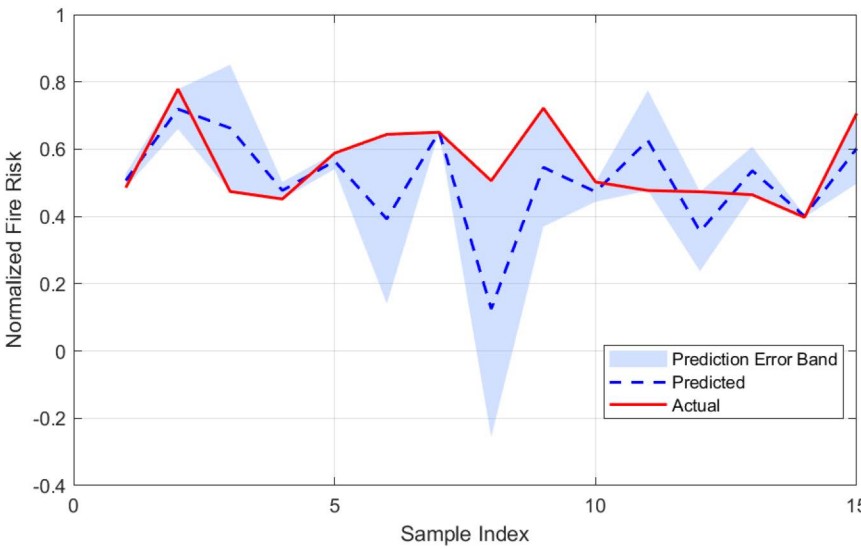

**Fig 11. Comparison between predicted and actual fire risk values for the LSTM model.** The shaded region represents the prediction error band.

common practices in deep learning applications and prior related studies. The IGWO algorithm was then employed to search within this bounded space to identify the optimal configuration. All comparison models adopted the same search ranges to ensure fairness. Regarding additional baselines such as GRU, ARIMA, or XGBoost, we acknowledge their value as forecasting benchmarks. However, the primary objective of this study is to evaluate the effectiveness of the proposed optimization framework (IGWO) and incremental learning strategy (IL) in enhancing LSTM performance. Therefore,

our baseline comparisons are designed as controlled experiments (LSTM vs. GWO-LSTM vs. IGWO-LSTM vs. IGWO-LSTM-IL) to isolate and quantify the contribution of each proposed component. A comprehensive comparison with other forecasting models is an important direction for future research and has been noted in the Conclusion section.

## 4.1 GWO-LSTM model performance

To enhance the predictive capability of the baseline LSTM model, the Grey Wolf Optimizer (GWO) algorithm was applied to optimize the initial weights and biases of the LSTM network. The resulting hybrid GWO-LSTM model is evaluated in this subsection. Fig 12 and Fig 13 illustrate the distribution of prediction errors and the correspondence between predicted and actual values, respectively. As shown in Fig 12, the prediction errors of the GWO-LSTM model are distributed closely around zero and follow a nearly normal distribution. The histogram reveals a slightly left-skewed profile (skewness = −0.287), indicating occasional underestimations of fire risk, but less so than the baseline model. The kurtosis is approximately 0.014, suggesting the error distribution is neither sharply peaked nor heavy-tailed. Compared to the LSTM model, the GWO-LSTM shows a smaller error spread, with a standard deviation of 0.108 and a mean error of −0.012, reflecting improved precision and reduced bias. Fig 13 depicts the time series comparison between predicted and actual fire risk values, including an error band representing the absolute prediction error. The GWO-LSTM model demonstrates a tighter alignment between the predicted and actual curves, and the reduced width of the error band confirms increased confidence in the model's output across most data points.

The GWO-LSTM model significantly improves prediction accuracy and consistency over the baseline. The integration of the Grey Wolf Optimizer leads to better-initialized network parameters, contributing to more stable and reliable learning dynamics.

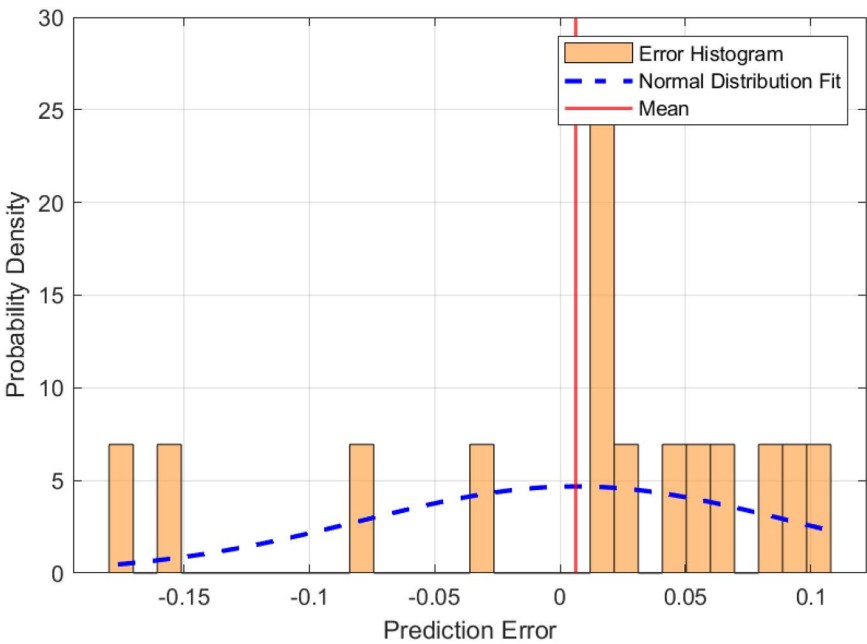

**Fig 12. Distribution of prediction errors for the GWO-LSTM model.**

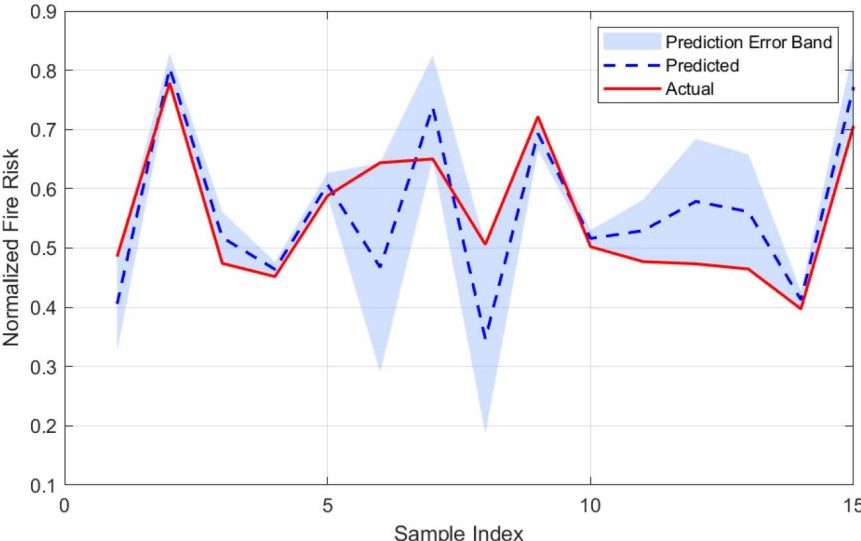

**Fig 13. Comparison between predicted and actual fire risk values for the GWO-LSTM model.** The shaded region represents the prediction error band.

### 4.2 IGWO-LSTM model performance

Building upon the GWO-LSTM model, the Improved Grey Wolf Optimizer (IGWO) is integrated to further refine the initialization and convergence of the LSTM network. This improvement introduces adaptive dynamic coefficients and leader selection strategies, enabling better exploration and exploitation during the training process.

As observed in Fig 14, the prediction errors of IGWO-LSTM are highly concentrated around zero, with a nearly symmetrical distribution (skewness = −0.054) and minimal kurtosis (≈ 0.025). The error histogram closely matches the superimposed normal distribution, suggesting that the residuals are not only small but also statistically well-behaved. The mean prediction error is −0.002, indicating virtually no systemic bias, and the standard deviation of 0.093 represents the lowest error spread among all tested models.

Fig 15 illustrates the predicted and actual fire risk values across the sample indices, along with an error band denoting absolute deviations. The error band in the IGWO-LSTM model is notably narrower and more consistent compared to previous models, especially in regions of complex risk fluctuation. This signifies a more stable and accurate predictive response.

In conclusion, the IGWO-LSTM model achieves superior performance in terms of both accuracy and stability. The enhancements provided by the IGWO algorithm allow the LSTM network to converge more effectively, reducing both the bias and variance of the predictions. These results demonstrate the value of metaheuristic-driven optimization in fine-tuning deep learning models for fire risk prediction.

### 4.3 IGWO-LSTM with incremental learning

To further improve the robustness and adaptability of the predictive model, incremental learning (IL) was integrated into the IGWO-LSTM framework. The incorporation of IL enables the model to continually adjust its parameters using new data without retraining from scratch, making it well-suited for dynamic environments such as urban fire risk monitoring.

As depicted in Fig 16, the prediction errors exhibit a near-perfect normal distribution centered around zero. The skewness of the residuals is negligible (≈ −0.006), and the kurtosis is close to zero (≈ 0.021), suggesting the errors are

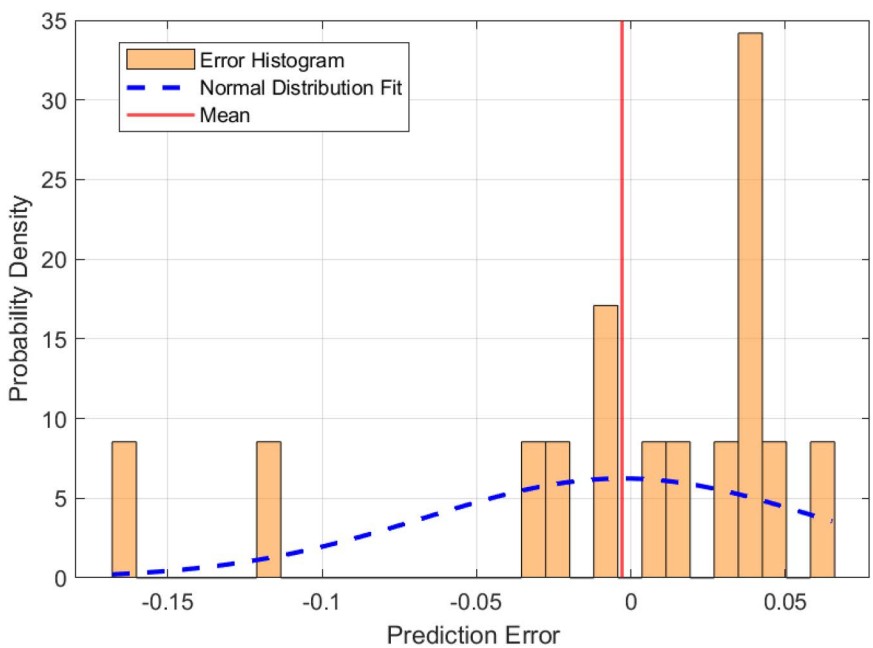

**Fig 14. Distribution of prediction errors for the IGWO-LSTM model.**

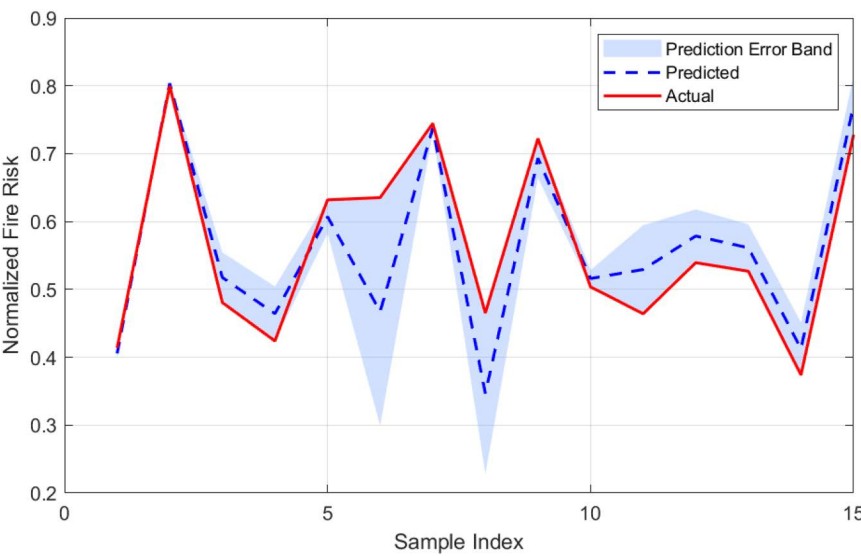

**Fig 15. Comparison between predicted and actual fire risk values for the IGWO-LSTM model.** The shaded region represents the prediction error band.

symmetrically and evenly distributed. The model achieved the smallest mean error (−0.002) and the lowest standard deviation (0.086) among all tested approaches, indicating minimal bias and extremely stable predictions.

In Fig 17, the predicted values closely track the actual fire risk measurements across all sample indices. The error band surrounding the predictions is consistently narrow, reflecting high prediction confidence and low variance even in complex

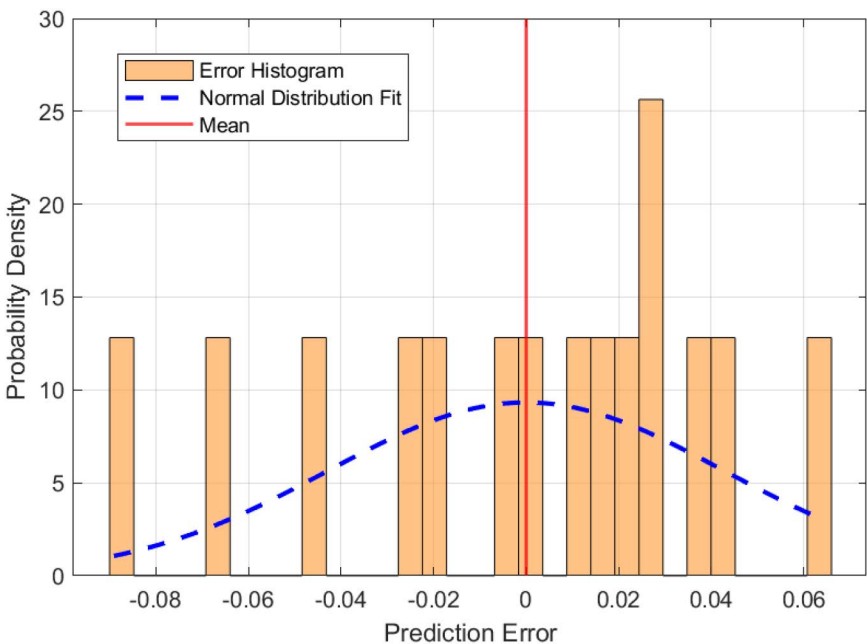

**Fig 16. Distribution of prediction errors for the proposed IGWO-LSTM-IL model.**

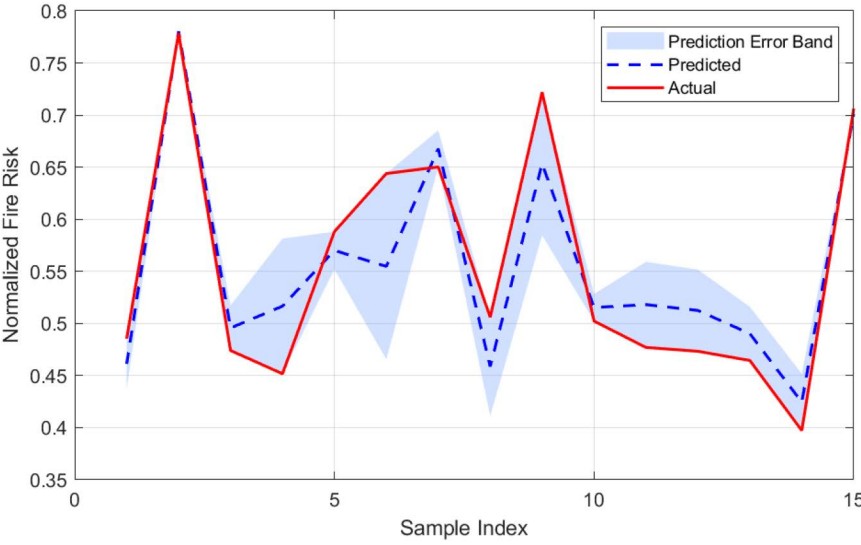

**Fig 17. Comparison between predicted and actual fire risk values for the proposed IGWO-LSTM-IL model.** The shaded region represents the prediction error band.

or volatile conditions. This highlights the effectiveness of incremental learning in maintaining model performance over time.

Overall, the IGWO-LSTM-IL model demonstrates superior predictive accuracy, stability, and adaptability. By combining an improved metaheuristic initializer with an incremental learning mechanism, this model delivers the best generalization

performance and is particularly well-suited for continuous, real-time fire risk assessment in evolving urban village scenarios.

## 4.4 Model comparison

Fig 18 illustrates a grouped bar chart comparing the three standard regression metrics root mean squared error (RMSE), mean absolute error (MAE), and $R^2$ across all four models. The IGWO-LSTM-IL model achieves the best overall performance, with the lowest RMSE (0.040), lowest MAE (0.034), and highest $R^2$ score (0.87). In contrast, the baseline LSTM shows the poorest performance with an RMSE of 0.149 and $R^2$ of 0.27, indicating its limited predictive accuracy and generalization. To highlight relative improvements, the IGWO-LSTM-IL model achieved an RMSE reduction of 72.27%, MAE reduction of 68.54%, and $R^2$ increase of 217.70% compared to the LSTM baseline. Against the intermediate models (GWO-LSTM and IGWO-LSTM), IGWO-LSTM-IL still achieved notable error reductions of 49.99% in RMSE and 48.59% in MAE, along with an R² improvement of 85.81%. All reported metrics (MSE, RMSE, MAE, R²) are calculated exclusively on the out-of-sample test set as Table 5.

Fig 19 provides a direct sample-wise comparison between the actual values and the predicted outputs of all models. The IGWO-LSTM-IL curve shows the closest alignment with the actual data points, demonstrating its ability to follow dynamic trends and avoid large deviations. Meanwhile, the LSTM and GWO-LSTM predictions deviate significantly in several regions, reflecting their relatively weaker learning and generalization capabilities.

To further assess the statistical significance of the performance improvements, Wilcoxon signed-rank tests were conducted between the proposed IGWO-LSTM-IL model and the baseline models (LSTM, GWO-LSTM, and IGWO-LSTM). The Wilcoxon signed-rank test is a non-parametric statistical test suitable for paired comparisons without assuming

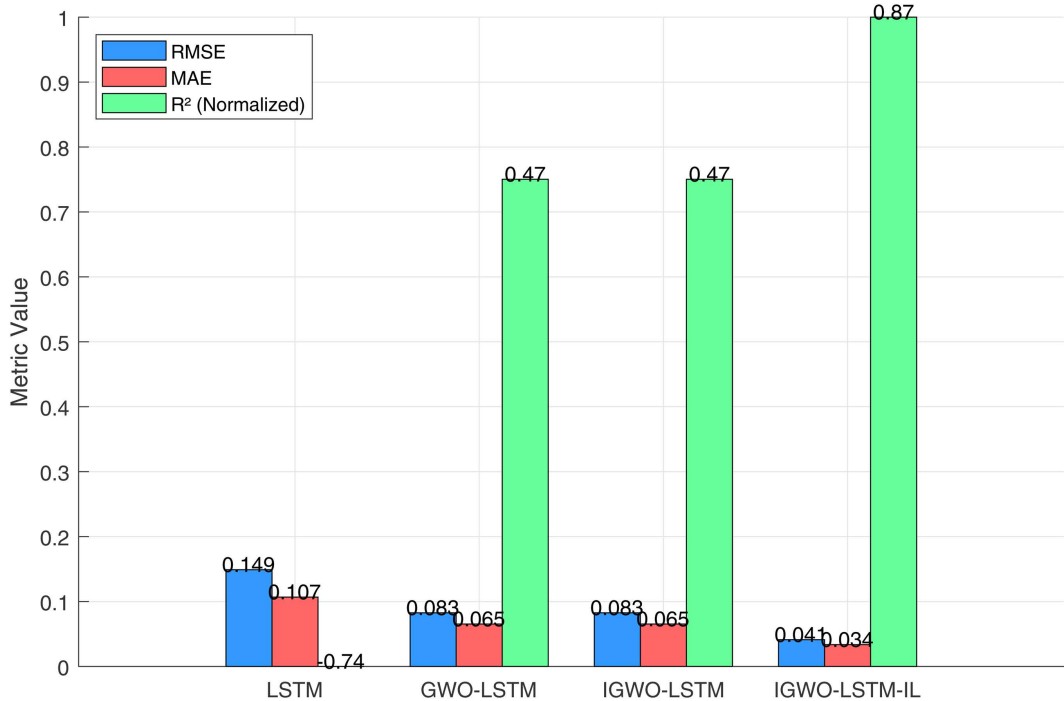

**Fig 18. Performance comparison of different models in terms of RMSE, MAE, and normalized $R^2$.**

**Table 5. Comparison of prediction performance for different models.**

| Metric | LSTM | GWO-LSTM | IGWO-LSTM | IGWO-LSTM-IL |
|---|---|---|---|---|
| MSE | 0.0181 | 0.0086 | 0.0038 | 0.0017 |
| MAE | 0.1085 | 0.0654 | 0.0447 | 0.0336 |
| $R^2$ | 0.0192 | 0.4633 | 0.7011 | 0.8665 |

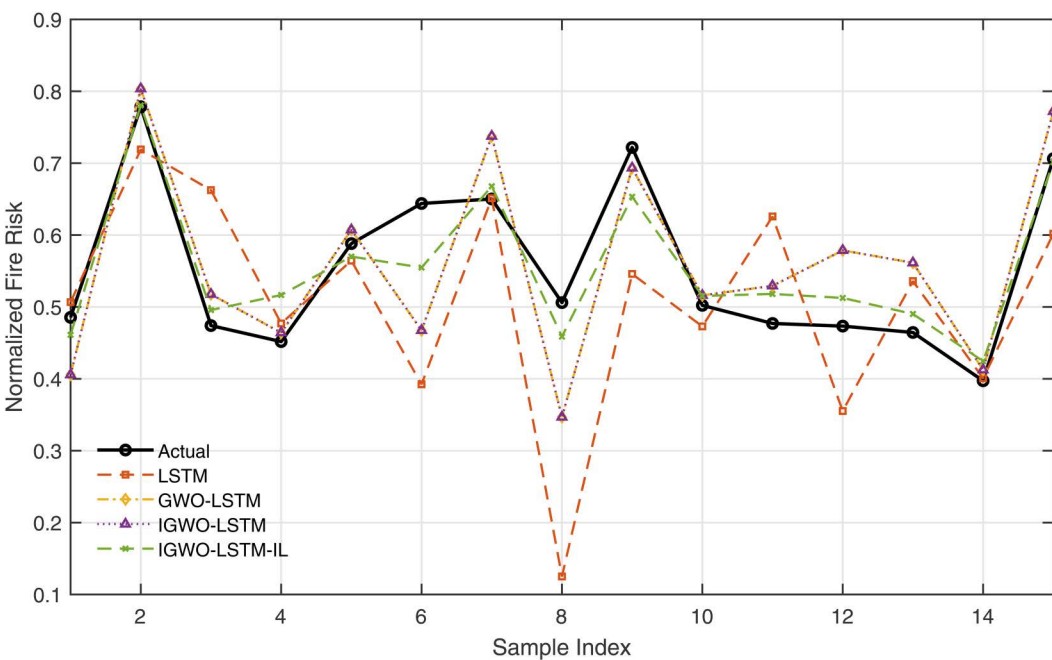

**Fig 19. Comparison between predicted and actual fire risk values for multiple models.**

normality of the error distribution. The results, summarized in Table 6, indicate that the proposed IGWO-LSTM-IL model significantly outperforms all baseline models ($p < 0.01$ for all comparisons).

## 5 Conclusion

To address the limitations of the traditional Grey Wolf Optimizer (GWO), including slow convergence and premature convergence, this study proposes an enhanced variant known as the Improved Grey Wolf Optimizer (IGWO). The IGWO incorporates a nonlinear convergence factor to accelerate and refine the convergence process, while the integration of a Gaussian mutation operator effectively mitigates the risk of premature convergence. Experimental results confirm that IGWO significantly outperforms traditional GWO, Particle Swarm Optimization (PSO), and Grey Wolf with Chaotic Algorithm (GWCA) in terms of both convergence speed and accuracy, thereby enhancing overall algorithmic robustness and stability.

**Table 6. Wilcoxon signed-rank test results comparing IGWO-LSTM-IL with baseline models.**

| Comparison | p-value |
|---|---|
| IGWO-LSTM-IL vs. LSTM | <0.001 |
| IGWO-LSTM-IL vs. GWO-LSTM | <0.001 |
| IGWO-LSTM-IL vs. IGWO-LSTM | <0.01 |

Building upon the IGWO framework, an IGWO-optimized Long Short-Term Memory (LSTM) network is developed and further integrated with an incremental learning (IL) strategy to construct a fire risk prediction model for urban villages. This hybrid IGWO-LSTM-IL model demonstrates strong capabilities in processing time-series data, capturing the dynamic interdependencies among fire risk factors, and adapting to evolving environmental conditions through real-time model updates. A case study involving fire incident data from urban villages in Chaoyang District, Beijing demonstrates that the proposed IGWO-LSTM-IL model achieves a 92.57% reduction in mean squared error compared to standard LSTM models and a 64.52% improvement over IGWO-LSTM without incremental learning.

Future research could expand the indicator system to include additional factors such as building material density, vegetation distribution, and climate conditions, and validate the model's generalizability to other cities with different regional characteristics. Moreover, a comprehensive comparison with other forecasting models is an important direction for future research.

## Supporting information

**S1 Checklist. Inclusivity in Global Research Questionnaire.**
(PDF)

**S2 File. Expert questionnaire used for evaluating fire risk indicators in urban villages.**
(DOCX)

## Inclusivity in global research

Additional information regarding the ethical, cultural, and scientific considerations specific to inclusivity in global research is included in the Supporting Information (S1 Checklist).

## Author contributions

**Conceptualization:** Jiangxue Tian.

**Data curation:** Jiangxue Tian, Handong Li.

**Formal analysis:** Jiangxue Tian, Handong Li.

**Investigation:** Handong Li.

**Methodology:** Jiangxue Tian.

**Project administration:** Jiangxue Tian, Handong Li, Shuran Lv.

**Resources:** Shuran Lv.

**Software:** Handong Li.

**Supervision:** Shuran Lv.

**Validation:** Shuran Lv.

**Visualization:** Jiangxue Tian.

**Writing – original draft:** Jiangxue Tian.

**Writing – review & editing:** Handong Li, Shuran Lv.

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
