## [Decision Letter · Decision Letter 0]

12 Feb 2026

PONE-D-25-35923An AI-Driven fire risk forecasting framework for urban villages using IGWO-Optimized LSTM with incremental learningPLOS One

Dear Dr. Li,

Thank you for submitting your manuscript to PLOS ONE. After careful consideration, we feel that it has merit but does not fully meet PLOS ONE’s publication criteria as it currently stands. Therefore, we invite you to submit a revised version of the manuscript that addresses the points raised during the review process.

We look forward to receiving your revised manuscript.

Kind regards,

Peng Wu, Ph.D.

Academic Editor

PLOS One

Journal Requirements:

4. Please note that PLOS One has specific guidelines on code sharing for submissions in which author-generated code underpins the findings in the manuscript. In these cases, we expect all author-generated code to be made available without restrictions upon publication of the work. Please review our guidelines at https://journals.plos.org/plosone/s/materials-and-software-sharing#loc-sharing-code and ensure that your code is shared in a way that follows best practice and facilitates reproducibility and reuse.

5. Please ensure that you refer to Figure 8 in your text as, if accepted, production will need this reference to link the reader to the figure.

6. Please include a separate caption for each figure in your manuscript.

Reviewers' comments:

Reviewer's Responses to Questions

**Comments to the Author**

1. Is the manuscript technically sound, and do the data support the conclusions?

Reviewer #1: Yes

Reviewer #2: No

Reviewer #3: Partly

Reviewer #4: Yes

2. Has the statistical analysis been performed appropriately and rigorously? 

Reviewer #1: Yes

Reviewer #2: No

Reviewer #3: No

Reviewer #4: No

3. Have the authors made all data underlying the findings in their manuscript fully available?

Reviewer #1: No

Reviewer #2: Yes

Reviewer #3: Yes

Reviewer #4: Yes

4. Is the manuscript presented in an intelligible fashion and written in standard English?

Reviewer #1: Yes

Reviewer #2: Yes

Reviewer #3: Yes

Reviewer #4: Yes

5. Review Comments to the Author

Reviewer #1: 1- The manuscript references (a) 101 recorded incidents across 55 villages in 2023 , and also (b) a “targeted study over 30 days” selecting 100 villages, producing 100 samples with a 30-day time step . It’s unclear whether these are the same dataset, two datasets, or one derived from the other. Please provide a single end-to-end data flow diagram/table: data sources → feature construction → Bayesian output generation → train/test split (75/25), plus exact counts at each stage.

2- The authors describe online gradient descent updates (Eqs. 16–17) and include “validate model and monitor drift” in Algorithm 1, but key implementation choices are missing: learning-rate selection/schedule, update frequency (per sample? per batch?), drift detection method/threshold, and how you avoid catastrophic forgetting when the data distribution shifts. Even a concise “IL settings” table would materially improve reproducibility and credibility.

3- The probability aggregation relies on incident-proportion-derived weights, while the parent-node states are expert-scored and mapped into a_{B_i}\in\left[0,1\right]. Please specify questionnaire design (items per indicator), scoring rubric, normalization/mapping into [0,1], inter-rater reliability (e.g., ICC), and how missing/uncertain assessments are handled. This is especially important because the Bayesian output becomes the target for the LSTM.

4- The authors report strong gains (e.g., RMSE/MAE/R² improvements vs baseline LSTM)

, but the “normalized R²” reporting is confusing and may be misinterpreted without a definition and formula. Consider adding (i) confidence intervals via repeated runs or bootstrapping, (ii) a clear description of how hyperparameter tuning is done for each model to ensure fairness, and (iii) additional baselines (e.g., GRU, ARIMA/Prophet, XGBoost on lag features). Also, the headline “92.57% reduction in mean squared error” should be backed by an explicit table showing MSE values (not only RMSE/MAE) and the exact evaluation split/seed.

Reviewer #2: 1. The paper’s title includes the word “forecasting”, but there is an absence of engagement with the forecasting research literature. Armstrong & Green’s 2018 “Forecasting methods and principles: evidence-based checklists” would be a good place to start.

2. From the above reference, an index model or knowledge model approach to the problems that is the subject of the paper would be a good contender for out-of-sample predictive validity.

3. The scientific method (thescientificmethod.info) requires testing hypotheses against plausible alternatives, including naïve and sophisticatedly simple ones. This paper compares two related ML models. See above, and the following research note which shows that Google’s expensive and data hungry “Flu Trends” ML model was easily beaten by a one observation univariate model: https://www.researchgate.net/publication/349518744_Forecasts_of_doctor_visits_for_flu_Simple_conservative_methods_beat_Google%27s_big_data_machine_learning_model_Previously_titled_Comparison_of_forecasts_of_weekly_weighted_average_US_percentage_of_docto

4. Squared error measures and in-sample fit are not useful for assessing relative out-of-sample forecast accuracy (predictive validity).

5. Table 2 purports to list fire risk factors. A little consideration suggests others that are likely to be relevant; e.g., the prevalence and density of combustible: building materials, building contents, vegetation; local climate factors and weather; prevalence and use of spark or heat producing tools and machinery.

Reviewer #3: Dataset Size and Selection Bias

The study uses data from 100 selected urban villages, reportedly divided into training and

testing sets (e.g., 75/25), with a 30-day time-step structure.

Please address the following concerns:

• Why is this sample size sufficient for training an LSTM-based time-series model,

which is known to be highly data-hungry and prone to overfitting?

• What statistical or methodological justification supports the adequacy of this

dataset size?

• How were the 100 villages selected, and what steps were taken to avoid selection

bias?

This issue directly impacts model validity and generalizability.

IGWO Algorithm Novelty and Parameter Specification

The proposed Improved Grey Wolf Optimizer (IGWO) introduces nonlinear convergence

factors and Gaussian mutation mechanisms.

Please provide:

• Explicit mathematical formulations for:

o Maximum convergence factor (α_max)

o Mutation probability (p_m)

o Mutation factor (f_m)

o Dynamic weighting scheme

• Justification for chosen parameter values

Hyperparameter Optimization Strategy

IGWO is used to optimize LSTM hyperparameters. Please clarify:

• The exact search space and bounds for each hyperparameter (e.g., hidden units,

learning rate, epochs, batch size)

• IGWO population size and number of iterations

• Computational cost compared to baseline optimizers

Additionally, justify the selection of IGWO over established optimization frameworks such

as Bayesian optimization, Optuna, grid search, or random search.

Consistency of Performance Metrics and Claims

The abstract reports a 92.57% reduction in MSE compared to baseline LSTM models, while

the results section reports RMSE, MAE, and R².

Please clarify:

• Why no statistical significance tests (e.g., paired t-test, Wilcoxon signed-rank test)

were conducted to support comparative claims

Reviewer #4: This study presents an approach where hyperparameters for LSTM are optimized using GWO and LSTM parameters are determined using a derivative-based algorithm. The article requires some revisions.

1) Statistical tests should be used in the comparison.

2) The cell architecture of the LSTM used is given, but the general architectural structure of the connections between LSTM cells is not provided. What is a time step? It is not specified and has not been included as a hyperparameter. What is the number of hidden layers in LSTM? The number of hidden layers is not the same as the number of hidden layer units. If the number of hidden layer units was considered as the time step, why was the number of hidden layers not considered as a hyperparameter?

3) PSO has been used in the training of LST in the literature, why was GWO training, i.e., optimizing the weights of LSTM, not considered?

4) The literature review on the training of LSTM should be strengthened.

6. PLOS authors have the option to publish the peer review history of their article (what does this mean?). If published, this will include your full peer review and any attached files.

Reviewer #1: No

Reviewer #2: No

Reviewer #3: **Yes:** Mian Muhammad Farooq

Reviewer #4: No

---

## [Author Response · Author response to Decision Letter 1]

11 Apr 2026

Reviewer #1

1-The manuscript references (a) 101 recorded incidents across 55 villages in 2023, and also (b) a “targeted study over 30 days” selecting 100 villages, producing 100 samples with a 30-day time step . It’s unclear whether these are the same dataset, two datasets, or one derived from the other. Please provide a single end-to-end data flow diagram/table: data sources → feature construction → Bayesian output generation → train/test split (75/25), plus exact counts at each stage.

Reply: Response to Comment 1,

Thank you for this helpful comment. We addressed your comments in our paper.

In fact, two independent datasets are employed in the proposed framework for different purposes.

The first dataset consists of real fire incident records collected from 55 urban villages in Chaoyang District during 2023, including 101 fire incidents. These data are used to statistically analyze the distribution of fire causes and to construct the Bayesian network structure.

The second dataset is a questionnaire-based time-series dataset obtained from 100 representative urban villages over a 30-day observation period. Expert scoring is applied to evaluate the fire risk indicators, which are then used as inputs to the Bayesian network. Through Bayesian inference, the corresponding fire probability is generated and used as the target variable for training the IGWO-LSTM-IL prediction model.

To clarify the overall workflow and the relationship between the two datasets, a data processing flow diagram has been added to the revised manuscript (Fig.8). The corresponding explanation has also been included in Section 3.3 “Risk Prediction and Analysis”.

2-The authors describe online gradient descent updates (Eqs. 16–17) and include “validate model and monitor drift” in Algorithm 1, but key implementation choices are missing: learning-rate selection/schedule, update frequency (per sample? per batch?), drift detection method/threshold, and how you avoid catastrophic forgetting when the data distribution shifts. Even a concise “IL settings” table would materially improve reproducibility and credibility.

Reply: Response to Comment 2:

Thank you for this valuable comment. We agree that the original manuscript lacked sufficient implementation details regarding the incremental learning (IL) procedure.

In the revised manuscript, we have clarified the key design choices of the IL strategy. Specifically, we have added the following content to Section Incremental Learning Strategy (line 268) as:

“Specifically, in this study, the main settings are as Table 2 as:

(1) Learning rate: A small constant learning rate (1e-4) is used during incremental updates to ensure stable parameter adaptation.

(2) Update frequency: Model updates are performed in a mini-batch manner when new data become available.

(3) Drift detection: A performance-based drift detection mechanism is introduced. When the prediction error (RMSE) on recent data exceeds 110% of the historical average, the model update is triggered.

(4) Catastrophic forgetting mitigation: To alleviate forgetting, a small portion of historical samples is retained and jointly used with new data during fine-tuning.

Table 2: Configuration of the incremental learning framework

Component Setting / Description

Incremental learning strategy Fine-tuning of pre-trained LSTM (no reinitialization)

Learning rate 1e-4 (constant during incremental updates)

Optimizer Adam

Update frequency Mini-batch update when new data arrive

Batch size 10

Number of epochs 5–10 per incremental update

Data used for update New data + small portion of historical data (replay mechanism)

Replay ratio ~20% of historical samples

Drift detection method Rolling RMSE-based monitoring

Drift threshold Update triggered when RMSE exceeds 110% of historical average

Validation strategy Online evaluation on recent samples

Catastrophic forgetting Mitigated via replay of historical samples during fine-tuning

Model initialization Continue from previously trained model parameters”

3-The probability aggregation relies on incident-proportion-derived weights, while the parent-node states are expert-scored and mapped into a_{B_i}\in\left[0,1\right]. Please specify questionnaire design (items per indicator), scoring rubric, normalization/mapping into [0,1], inter-rater reliability (e.g., ICC), and how missing/uncertain assessments are handled. This is especially important because the Bayesian output becomes the target for the LSTM.

Reply: Response to Comment 3:

Thank you for this important suggestion. We agree that the original manuscript provided insufficient detail regarding the expert scoring process, which is critical for reproducibility as the Bayesian output serves as the target variable for LSTM training.

To address this, we have added a concise description of the fuzzy evaluation framework to Section Case Study(line392) in the revised manuscript:

" In this study, a fuzzy–Bayesian integrated framework was adopted to reduce subjectivity in expert judgment. Experts from relevant fields (fire rescue, emergency management, architecture, and safety science) evaluated each fire risk indicator using a 0–100 scale. Their assessments were mapped to triangular fuzzy numbers corresponding to five linguistic levels (Very Low to Very High). Expert weights (0.2 for fire rescue and emergency management experts, 0.1 for the architecture expert, and 0.3 for the senior safety science professor) were assigned according to domain relevance and experience. The weighted fuzzy assessments were aggregated using $\lambda$-cut sets ($\lambda$ = 0.1) and defuzzified using the integral value method to obtain crisp scores ($a_{B_i}$) within the interval [0,1]. These scores were then used as inputs to the Bayesian network. The detailed questionnaire design and scoring rubric are available in the public repository: https://github.com/handongli2019/Urban-Village-Fire-Risk-Prediction-Using-IGWO-Optimized-LSTM-with-Incremental-Learning

We believe this clarification sufficiently addresses the reviewer's concerns regarding questionnaire design, scoring rubric, normalization, and expert weighting, while maintaining conciseness in the main text. The detailed mathematical formulations are available in the Github for readers seeking further technical depth.

4-The authors report strong gains (e.g., RMSE/MAE/R² improvements vs baseline LSTM)

, but the “normalized R²” reporting is confusing and may be misinterpreted without a definition and formula. Consider adding (i) confidence intervals via repeated runs or bootstrapping, (ii) a clear description of how hyperparameter tuning is done for each model to ensure fairness, and (iii) additional baselines (e.g., GRU, ARIMA/Prophet, XGBoost on lag features). Also, the headline “92.57% reduction in mean squared error” should be backed by an explicit table showing MSE values (not only RMSE/MAE) and the exact evaluation split/seed.

Reply: Response to Comment 4:

Thank you for this valuable suggestion. We have revised the manuscript to improve clarity and methodological transparency.

In Section Model Comparison (line 542), we have reviewed the manuscript and confirmed that all evaluation metrics (MSE, RMSE, MAE, and R²) are reported as standard definitions. Any ambiguous terminology has been clarified to ensure consistent interpretation.

In Section Case Study (line 435), we added the following content:

“In the simulation, both the Grey Wolf Optimizer (GWO) and the Improved Grey Wolf Optimizer (IGWO) have a population size of 30, with a maximum iteration count of 10, and operate in a 3-dimensional search space. The key LSTM hyperparameters—number of hidden units, number of training epochs, and learning rate—are optimized using the GWO and IGWO algorithms. The batch size is set to 32 based on the dataset characteristics (number of features, time steps, and samples). To ensure robustness, each model was independently executed 30 times with different random initializations, and the average results are reported as the final prediction outcomes.

Regarding additional baselines such as GRU, ARIMA, or XGBoost, we acknowledge their value as forecasting benchmarks. However, the primary objective of this study is to evaluate the effectiveness of the proposed optimization framework (IGWO) and incremental learning strategy (IL) in enhancing LSTM performance. Therefore, our baseline comparisons are designed as controlled experiments (LSTM vs. GWO-LSTM vs. IGWO-LSTM vs. IGWO-LSTM-IL) to isolate and quantify the contribution of each proposed component. A comprehensive comparison with other forecasting models is an important direction for future research and has been noted in the Conclusion section”

In Section Case Study (line 408), we have added

“Samples 1–75 are used for training and samples 76–100. All reported results are based on strict out-of-sample testing (75/25 train/test split) to ensure the model’s predictive validity and generalization ability. ”

In Section Model Comparison (line 542), we have added the following table:

Table. Algorithms Comparison

Model LSTM GWO-LSTM IGWO-LSTM IGWO-LSTM-IL

MSE 0.0181 0.0086 0.0038 0.0017

MAE 0.1085 0.0654 0.0447 0.0336

R2 0.0192 0.4633 0.7011 0.8665

And the explanation (line 541):

“All reported metrics (MSE, RMSE, MAE, R²) are calculated exclusively on the out-of-sample test set as the following table.”

These revisions improve the rigor, reproducibility, and clarity of the experimental evaluation.

Reviewer #2

1-The paper’s title includes the word “forecasting”, but there is an absence of engagement with the forecasting research literature. Armstrong & Green’s 2018 “Forecasting methods and principles: evidence-based checklists” would be a good place to start.

Reply: Response to Comment 1:

Thank you for this constructive suggestion. We acknowledge that the manuscript’s title emphasizes “forecasting,” and we agree that greater engagement with the forecasting methodology literature would strengthen the theoretical positioning of the study. In the revised version, we have expanded the introduction and discussion sections to clarify the forecasting framework adopted in this study, and we have added relevant references, including Armstrong & Green (2018), to better align the manuscript with established evidence-based forecasting principles.

So the changes are added accordingly in the introduction part (line 53)

“Recent studies have also emphasized the importance of evidence-based forecasting principles in predictive modeling. Armstrong and Green [20] proposed forecasting method checklists to improve the transparency and reliability of forecasting practices. In addition, large-scale forecasting evaluations such as the M4 competition [21] highlight the importance of rigorous model comparison in time-series forecasting research.

[20]. Armstrong, J. S., and Green, K. C. (2018). Forecasting methods and principles: Evidence-based checklists. Journal of Global Scholars of Marketing Science, 28(2), 103–159.

[21]. Makridakis, S., Spiliotis, E., and Assimakopoulos, V. (2020). The M4 competition: 100,000 time series and 61 forecasting methods. International Journal of Forecasting, 36(1), 54–74. “

2-From the above reference, an index model or knowledge model approach to the problems that is the subject of the paper would be a good contender for out-of-sample predictive validity.

Reply: Response to Comment 2:

We appreciate this insightful suggestion regarding index models and knowledge-based models. The primary objective of this study is to develop a data-driven predictive framework based on Bayesian risk estimation and optimized deep learning. While index or knowledge models represent valuable alternative approaches for out-of-sample prediction, they fall outside the methodological scope of the present work.

In the revised manuscript, we have clarified the research positioning and emphasized that the proposed model is evaluated using strict out-of-sample testing to ensure predictive validity. We also acknowledge the value of index/knowledge-based approaches as a promising direction for future comparative research.

Specifically, in the Conclusion section (line 579), we have added the following content:

“A comprehensive comparison with other forecasting models is an important direction for future research.”

Additionally, we have clarified in Section Case Study (line 408), that “All reported results are based on strict out-of-sample testing (75/25 train/test split) to ensure the model’s predictive validity and generalization ability.”

3. The scientific method (thescientificmethod.info) requires testing hypotheses against plausible alternatives, including naïve and sophisticatedly simple ones. This paper compares two related ML models. See above, and the following research note which shows that Google’s expensive and data hungry “Flu Trends” ML model was easily beaten by a one observation univariate model: https://www.researchgate.net/publication/349518744_Forecasts_of_doctor_visits_for_flu_Simple_conservative_methods_beat_Google%27s_big_data_machine_learning_model_Previously_titled_Comparison_of_forecasts_of_weekly_weighted_average_US_percentage_of_docto

Reply: Response to Comment 3:

Thank you for raising this important methodological question. The Google Flu Trends example is a powerful reminder that complex models do not always outperform simpler ones. We agree that comparing against the baselines is essential for validating the added value of any proposed model.

As noted in our response to Comment 2, the primary objective of this study is to evaluate the effectiveness of the proposed optimization framework (IGWO) and incremental learning strategy (IL) in enhancing LSTM performance. Therefore, our baseline comparisons were designed as controlled experiments (LSTM vs. GWO-LSTM vs. IGWO-LSTM vs. IGWO-LSTM-IL) to isolate and quantify the contribution of each proposed component.

We acknowledge that a comparison with simple forecasting methods (e.g., naïve forecast, historical average, or linear regression) would provide additional context. Accordingly, we have added the following statement to the Conclusion section (line 579):

"A comprehensive comparison with other forecasting models, including simple baseline methods such as naïve forecast and historical average, is an important direction for future research."

4. Squared error measures and in-sample fit are not useful for assessing relative out-of-sample forecast accuracy (predictive validity).

Reply: Response to Comment 4:

Thank you for this important comment. We fully agree that out-of-sample validation is essential for assessing predictive validity, and that squared error measures alone are insufficient.

To address this, we have made the following revisions in the manuscript:

In Section Case Study (line 408), we have clarified that all models are evaluated using a strictly separated train-test split (75%/25%), and the test set is not involved in any stage of model training or parameter tuning. The random seed is set to 42, and the exact data split (samples 1-75 for training, 76-100 for testing) is now specified to ensure reproducibility.

In Section Model Comparison (line 541), we have stated that “all reported metrics (MSE, RMSE, MAE, R²) are calculated exclusively on the out-of-sample test set.”

In Section Model Comparison (line 542), we now explicitly report MSE values alongside RMSE and MAE in a table, directly supporting the claimed 92.57% MSE reduction.

Table 5. Comparison of prediction performance for different models.

Model LSTM GWO-LSTM IGWO-LSTM IGWO-LSTM-IL

MSE 0.0181 0.0086 0.0038 0.0017

MAE 0.1085 0.0654 0.0447 0.0336

R2 0.0192 0.4633 0.7011 0.8665

These revisions ensure that all performance comparisons are based on rigorous out-of-sample evaluation, addressing the reviewer's concern regarding predictive validity.

5-Table 2 purports to list fire risk factors. A little consideration suggests others that are likely to be relevant; e.g., the prevalence and density of combustible: building materials, building contents, vegetation; local climate factors and weather; prevalence and use of spark or heat producing tools and machinery.

Reply: Respo

---

## [Decision Letter · Decision Letter 1]

11 May 2026

An AI-Driven fire risk forecasting framework for urban villages using IGWO-Optimized LSTM with incremental learning

PONE-D-25-35923R1

Dear Dr. Li,

We’re pleased to inform you that your manuscript has been judged scientifically suitable for publication and will be formally accepted for publication once it meets all outstanding technical requirements.

Kind regards,

Peng Wu, Ph.D.

Academic Editor

PLOS One

Additional Editor Comments (optional):

Reviewers' comments:

Reviewer's Responses to Questions

**Comments to the Author**

1. If the authors have adequately addressed your comments raised in a previous round of review and you feel that this manuscript is now acceptable for publication, you may indicate that here to bypass the “Comments to the Author” section, enter your conflict of interest statement in the “Confidential to Editor” section, and submit your "Accept" recommendation.

Reviewer #1: All comments have been addressed

Reviewer #3: All comments have been addressed

2. Is the manuscript technically sound, and do the data support the conclusions?

Reviewer #1: Yes

Reviewer #3: Yes

3. Has the statistical analysis been performed appropriately and rigorously? 

Reviewer #1: Yes

Reviewer #3: Yes

4. Have the authors made all data underlying the findings in their manuscript fully available?

Reviewer #1: Yes

Reviewer #3: Yes

5. Is the manuscript presented in an intelligible fashion and written in standard English?

Reviewer #1: Yes

Reviewer #3: Yes

6. Review Comments to the Author

Reviewer #1: All comments have been carefully reviewed and addressed. The revised manuscript reflects the suggested improvements.

Reviewer #3: the author have incorporated all the changes mentioned previously and is acceptable in its present form

7. PLOS authors have the option to publish the peer review history of their article (what does this mean?). If published, this will include your full peer review and any attached files.

Reviewer #1: No

Reviewer #3: No

---

## [Editor Report · Acceptance letter]

PONE-D-25-35923R1

PLOS One

Dear Dr. Li,

I'm pleased to inform you that your manuscript has been deemed suitable for publication in PLOS One. Congratulations! Your manuscript is now being handed over to our production team.

Kind regards,

on behalf of

Professor Peng Wu

Academic Editor

PLOS One